# Opportunities of Integrating Slow Pyrolysis and Chemical Leaching for Extraction of Critical Raw Materials from Sewage Sludge

Andrea Salimbeni [1,2,*] , Marta Di Bianca [2,3] , Giacomo Lombardi [1,2] , Andrea Maria Rizzo [1,2] and David Chiaramonti [2,4]

1 Department of Industrial Engineering, University of Florence, 50139 Firenze, Italy
2 RE-CORD (Renewable Energy Consortium for Research and Demonstration), 50038 Scarperia e San Piero, Italy; marta.dibianca@re-cord.org (M.D.B.)
3 Department of Civil and Environmental Engineering, University of Florence, 50139 Firenze, Italy
4 "Galileo Ferraris" Energy Department, Polytechnic of Turin, 10129 Torino, Italy
* Correspondence: andrea.salimbeni@unifi.it

**Abstract:** Slow pyrolysis is a promising technology to convert sewage sludge into char: a stable solid product with high carbon and phosphorus content. However, due to its heavy metals content, char use in agriculture is avoided in many European Union (EU) countries. This study aimed to test a solution, based on integrating slow pyrolysis and chemical leaching, to separate phosphorus and other inorganics from char, obtaining an inorganic P-rich fertiliser and a C-rich solid usable for industrial purposes. The sludge was first characterized and then processed in a 3 kg/h slow pyrolysis reactor at 450 °C for 30 min. The resulting char was processed by chemical leaching with acid (HCl, $HNO_3$) and alkali (KOH) reagents to extract inorganic compounds. To optimize the inorganic extraction, three case studies have been considered. The char obtained from sewage sludge pyrolysis contained around 78% d.b. (dry basis) of inorganics, 14% d.b. of C, 14% d.b. of Al, and almost 5% d.b. of P. The leaching tests enabled to extract 100% of P, Mg, and Ca from the char. The remaining char contained mainly carbon (27%) and silica (42%), with a surface area of up to 70 $m^2$/g, usable as adsorbent or precursor of sustainable materials.

**Keywords:** pyrolysis; leaching; sewage sludge; phosphorous; critical raw materials; carbon; biocoal





## 1. Introduction

### 1.1. State of the Art of Sludge Disposal

Sewage sludge, the main by-product originated from wastewater treatment plants (WWTPs), is generated by primary sedimentation of wastewater or biological treatment and usually treated by thickening, stabilization by anaerobic digestion (with generation of biogas) or aerobic digestion, dewatering (to reach a 15–25% solid content), and eventual drying before its final disposal [1].

Chemically, sludge is generally composed of an organic fraction of microbial origin, inorganic fertilising compounds (phosphorous, potassium, iron, magnesium), other inorganic elements (silicon, calcium, aluminium), and hazardous compounds for the environment and human health (pathogenic microorganisms, persistent organic pollutants, other heavy metals) [1]. For this reason, management of sewage sludge has always represented one of the main criticalities linked to wastewater treatment, especially in the European Union (EU) [2], due to the high costs associated with sludge disposal (which can represent up to 50% of the current operating costs of a WWTP [3]) and high amount of sludge produced. In fact, in EU-28, the production of urban sewage sludge estimated by Eurostat is more than 10 Mt dry matter, of which around 55–65% is accounted to Germany, the UK, Spain, France, and Italy [4]. According to Eurostat, in EU-28, agriculture is the main sludge disposal route

(50%), followed by incineration (28%) and landfilling (18%), while the remaining fraction is disposed by different methods [4]. When sludge use on soil is not allowed, sludge producers are obliged to rely on incineration or landfilling, which usually have higher associated costs per ton of dry sludge compared to agricultural reuse [5]. Due to the high ash content and reduced calorific value of dry sludge, incineration represents an expensive disposal system for sludge producers [6,7]. Last, landfilling, due to its high environmental impact, is discouraged by the waste management hierarchy of Directive 2008/98/EC (amended by Directive 2018/851/EC [8]). However, a new circular approach towards waste management, also promoted by the Circular Economy Package and aimed at a transition to a circular economy [9], suggests that new routes for sewage sludge stabilization and conversion systems must be found.

In this context, sewage sludge could represent a valuable source of biogenic carbon and raw materials, such as phosphorus, aluminium, and silicon, of strategic importance for our economy. The key challenge is to identify innovative technologies that enable stabilization and use of sewage sludge as a resource from which to extract valuable products in order to contribute to development of a sustainable circular bioeconomy system.

*1.2. Slow Pyrolysis*

Pyrolysis is a thermochemical process by which the organic fraction of a feedstock is decomposed in the absence of oxygen at temperatures between 300 and 900 °C [10,11]. Compared to the combustion process, it has been reported that pyrolysis processing generates fewer air pollutant emissions [11], smaller amount of flue gases, acidic gases, and dioxins [12], and low nitrogen oxide and sulphur oxide are formed [13]. Moreover, pyrolysis eliminates pathogens [12], concurrently stabilizing organic matter and facilitating recovery of valuable elements (P, C, etc.). Sewage sludge needs to be dried to reach a moisture content under 30% to make it suitable for pyrolysis [13]. The pyrolysis process always generates a solid carbonaceous matrix (char), mixture of condensable gases (water and organic compounds), and mixture of permanent gases ($CO$, $CO_2$, $CH_4$, $H_2$, etc.) [14]. Variation in pyrolysis process parameters (temperature, heating rate, solid retention time) enables changing the composition and mass yield of the three different end-products [11].

When the pyrolysis process is applied to maximize production of char—and thus recovery of carbon in the solid product—it is commonly known as "slow pyrolysis". Slow pyrolysis has char as the main product [10] and involves relatively low heating rates (0.5–10 °C/min), temperatures from 400 to 600 °C, and long solids residence times (in the order of hours) [10,15]. Char from sludge pyrolysis is a carbon-rich material, hydrophobic, with low volatile content and relevant porosity [16]. Due to its properties, char can be activated to be used as an adsorbent, growing medium, or applied as soil amendment [12]. Phosphorous, as other inorganic elements, is concentrated in char after sludge pyrolysis, indicating that it is associated to the inorganic fraction of char [17]. Since most of the carbon contained in char is recalcitrant, its use on soil might enable sequestering carbon [18]. However, the literature reports that sewage sludge ash content ranges from 55.8 to 61.3%, resulting in a char ash content of 64.1–79.1% [17]. In fact, during pyrolysis, part of the feedstock organic matter is volatilized into pyrogas, while ashes are inert to the process and remain in the solid product. For this reason, char yields obtained by sludge pyrolysis are higher than lignocellulosic biomass feedstocks, usually around 20–35% [15]. Consequently, an effect of pyrolysis is accumulation of sludge heavy metals in the produced char [12], with potential harmful effects if used for soil application.

For this reason, despite the high concentration of phosphorus and other nutrients, application of sludge-derived char as a soil conditioner is limited by regulation (EU) 2019/1009 on the market of EU fertilizing products [19], as well as in many EU countries.

Moreover, due to its high ash content and thus low calorific value, char from sludge slow pyrolysis is neither suitable as a solid biofuel nor for use in the steel industry despite having high carbon content [20].

However, high concentrations of both inorganic elements and carbon make sludge-derived char an interesting raw material, usable as a source of nutrients and renewable carbon [21]. To this aim, a chemical upgrading step to extract phosphorus and other inorganic compounds could represent not only a promising solution to ensure full valorisation of phosphorus and other valuable inorganic compounds but also an opportunity to improve the char quality and unlock its potential application in cement, steel, and other industry sectors.

### 1.3. Chemical Leaching

Chemical leaching is a process that enables separating the soluble components of a solid material by dissolving them in a liquid phase [22]. A common application of chemical leaching is low-grade coal cleaning, aiming to reduce the amount of inorganic mineral matter [23], which comprises ash and sulphur [24]. Chemical leaching can be performed by use of acids or alkalis in one step or in stepwise processes or by a combination of both [24]. Chemical leaching by alkalis, such as NaOH, KOH, and Ca(OH)$_2$, is effective regarding extraction of silica, alumina, and clay-bearing minerals (which represent 60–90% of total coal mineral matter), producing hydrated alkali-bearing silicate, aluminate, and aluminosilicate complexes [23]. Inorganic and organic sulphur may be extracted by NaOH or KOH [25]. By acid leaching, carbonates, phosphates, Fe$_2$O$_3$, and sulphides are effectively extracted from low-grade coal, while clay-bearing minerals are not dissociated [23]. Acids can be applied sequentially to alkali since alkalis (such as NaOH and KOH) react with sulphur and the main coal minerals (silica, alumina, dolomite, quartz) to form hydrated alkali compounds of silicate and aluminate, which are then dissolved by acids, such as H$_2$SO$_4$ and HCl, with other unreacted minerals [23].

Acid leaching can find application on phosphorous recovery from wet, dewatered, or dry sewage sludge or from incinerated sludge ashes [26]. Extraction of phosphorus and other raw materials by chemical leaching of sludge-derived ashes was investigated in several studies. In fact, phosphorous can be recovered from ashes generated by sludge mono-incineration plants, for example, by wet leaching processes, which dissolve ashes from phosphorous, generally bonded to aluminium and calcium, in acidic environments using HCl or H$_2$SO$_4$ [26]. However, insufficient literature studies are available on chemical leaching of char from sludge slow pyrolysis; moreover, an assessment of chemical leaching efficiency should be provided considering not only extraction of phosphorus but also of other inorganic elements and the whole ash removal efficiency.

### 1.4. Case Study and Objective

A slow pyrolysis process enables to convert dry sludge into a porous solid with carbon in stable form, which facilitates leaching process efficacy [27]. To make char from sewage sludge usable, chemical leaching of sewage sludge char is an effective system to reduce char ash content and extract valuable mineral elements (such as phosphorous) in view of their recovery [28]. The process generates upgraded char with lower ash content and higher application potential and a liquid phase rich in mineral compounds, which can be further recovered. The challenge is to maximize extraction of phosphorus but also to optimize extraction of inorganic compounds, aiming to recover two products: the inorganic fraction and the upgraded char (the "biocoal"). The scope of this study is to assess the performances and opportunities of combining slow pyrolysis with a char chemical leaching process to separately recover phosphorus, aluminium, magnesium, and silicon from sludge, and also for upgrading sludge-derived char to one or more end-of-waste products. In particular, the present study aims at evaluating application of the pyrolysis-chemical leaching process to valorise the sewage sludge produced by an Italian WWTP located in Tuscany as an alternative route to the current disposal method.

## 2. Materials and Methods

### 2.1. Materials Characterization

Characterization of sewage sludge consisted of different physical–chemical analyses. Proximate analysis was aimed at definition of moisture content, ash content, determined at 550 °C (ash 550) and 710 °C (ash 710), volatiles content, and fixed carbon (fixed C), which was calculated as difference between 100 and the sum of moisture, volatiles, and ash 550. Ash and volatiles were determined by a thermogravimetric analyser (LECO TGA701). Ultimate analysis allowed to determine carbon, hydrogen, nitrogen, sulphur, and chlorine contents in the material, and it was conducted by a CHN-S analyser (LECO TruSpec CHN-S) and by use of a bomb calorimeter (LECO AC500) for pre-treatment and an ion chromatography system (Metrohm 883 Basic IC plus) to determine chlorine.

The higher heating value (HHV) of the feedstock was determined analytically by a bomb calorimeter (LECO AC500) and used by means of moisture and hydrogen content to derive its lower heating value (LHV). Characterization of the material was completed using microwave plasma atomic emission spectroscopy (MP-AES, by Agilent 4200 MP-AES) in order to quantify the concentration of metal oxides in the sewage sludge. In addition, energy dispersive X-ray spectroscopy (EDX, by Shimadzu EDX 7000) was employed to determine relative concentration of oxides in ashes of sewage sludge. These concentrations were multiplied by ash 710 to determine the absolute concentration of oxides in the material.

The liquids recovered from the two condensation units of the slow pyrolysis pilot plant were collected and mixed, and then the aqueous phase was separated from the organic phase by a separating funnel. To characterize the organic phase, its carbon, hydrogen, and nitrogen content (by a LECO TruSpec CHN-S analyser), its HHV (by a LECO AC500 bomb calorimeter), and its water content (by an automatic titrator Metrohm 848 Titrino Plus) were determined. The aqueous phase was dried to determine its dry matter content by means of a rotavapor (rotavapor IKA RV-10 Control). Based on the weight of the two phases, the overall composition of the recovered liquids was calculated.

Char and biocoal were characterized through the same analysis of the sewage sludge with the addition of surface area and pore size distribution, which were determined via BET surface area analyser (BET Quantachrome NOVA 2200E). Eluates were analysed by MP-AES to determine their composition in metals and other elements and calculate element extraction efficiency (EE). Oxides composition of biocoal from case 1 and case 2 was determined by EDX.

The analytical methods adopted for characterization of the sewage sludge and intermediate and final products are summarized in Table 1.

### 2.2. Composition of the Feedstock

The sewage sludge object of the study is produced by the secondary sedimentation phase after biological treatment, and then it is thickened, anaerobically digested, and finally dewatered. Use of aluminium polychloride (around 270 t/y; around 740 kg/d) and iron chloride (approximately 10 t/y; around 27 kg/d) in the plant as reagents results in a high inorganic matter content in the produced sludge. The physical–chemical characterization of the sewage sludge is reported in Table 2. The ash content represents almost 50% of the sludge dry matter.

**Table 1.** Analytical methods adopted for material characterisation.

| Parameter | Method |
|---|---|
| Moisture | UNI EN ISO 18134-2: 2017 [a] |
| Water content | ASTM E203-08 |
| Ash 550 | UNI EN ISO 18122: 2016 [a] |
| | UNI EN 13039: 2012 [b] |
| Ash 710 | UNI EN 1860-2: 2005 |
| Volatiles | UNI EN ISO 18123: 2016 |
| Fixed C | UNI EN 1860-2: 2005 |
| C, H, N | UNI EN ISO 16948: 2015 [a][b] |
| | ASTM D5291-10 [c] |
| S, Cl | UNI EN ISO 16994: 2017 |
| HHV | UNI EN ISO 18125: 2018 [a][b] |
| | DIN 51900-1:2000, DIN 51900-3:2005 [c] |
| LHV | UNI EN ISO 18125: 2018, UNI EN ISO 16948: 2015 [a][b] |
| | DIN 51900-1:2000, DIN 51900-3:2005, ASTM D5291-10 [c] |
| Surface area | ASTM D6556-10 |
| Metals | UNI EN ISO 16967: 2015, UNI EN ISO 16968: 2015 |

Notes: [a] for sewage sludge. [b] For char and biocoal. [c] For organic phase.

**Table 2.** Feedstock (sewage sludge) proximate, ultimate, and thermal analysis.

| Parameter | Value | Unit |
|---|---|---|
| Moisture | 73.9 | % w.b. |
| Ash 550 | 49.3 | % d.b. |
| Ash 710 | 48.5 | % d.b. |
| Volatiles | 46.0 | % d.b. |
| Fixed C | 4.7 | % d.b. |
| C | 23.6 | % d.b. |
| H | 4.0 | % d.b. |
| N | 3.8 | % d.b. |
| S | 0.8 | % d.b. |
| Cl | 0.1 | % d.b. |
| HHV | 10.3 | MJ/kg d.b. |
| LHV | 9.4 | MJ/kg d.b. |

Notes: w.b.—wet basis. d.b.—dry basis.

By the elemental composition of the sewage sludge (Table 3), we can observe high content of silicon (5.9%), aluminium (4.1%), phosphorous (3.8%), calcium (2.4%), and iron (1.6%). The elevated aluminium and iron content is due to aluminium polychloride and ferrous chloride dosage in the WWTP that originates the sewage sludge.

**Table 3.** Elemental composition of the feedstock (sewage sludge).

| Element | Value | Unit |
| --- | --- | --- |
| Al | 40,574 | mg/kg d.b. |
| B | 6 | mg/kg d.b. |
| Ba | 784 | mg/kg d.b. |
| Ca | 23,550 | mg/kg d.b. |
| Cr | 77 | mg/kg d.b. |
| Cu | 702 | mg/kg d.b. |
| Fe | 15,763 | mg/kg d.b. |
| K | 6055 | mg/kg d.b. |
| Li | 20 | mg/kg d.b. |
| Mg | 5792 | mg/kg d.b. |
| Mn | 358 | mg/kg d.b. |
| Na | 842 | mg/kg d.b. |
| Ni | 96 | mg/kg d.b. |
| P | 37,498 | mg/kg d.b. |
| Pb | 61 | mg/kg d.b. |
| Si | 58,947 | mg/kg d.b. |
| Ti | 277 | mg/kg d.b. |
| V | 34 | mg/kg d.b. |
| Zn | 453 | mg/kg d.b. |

Note: d.b.—dry basis.

### 2.3. Description of Slow Pyrolysis Pilot Unit

The slow pyrolysis test of sewage sludge was performed in a pilot plant designed and operated by RE-CORD, called SPYRO (Slow Pyrolysis ReactOr). The pilot plant is an auger type reactor with a maximum capacity of 3 kg/h of inlet feedstock (depending on the bulk density) and can be operated up to 600 °C. The solid feedstock is introduced into the reactor by a dosing screw through a double gate–valve system, which provides air-tightness for the process.

The solids residence time in the reactor can be varied from few minutes to 1 h by adjusting the rotating speed of the reactor screw, which transports the material from the inlet section to the final discharge as char in a sealed vessel for collection. The reactor is equipped with an auxiliary gas injection port section, which allows the entrance of $N_2$, maintaining an inert atmosphere. The reactor has three main independent heating sections, used to effectively control the heating rate and provide an extra degree of flexibility. Five principal thermocouples are located along the reactor's length to monitor the reactor temperature. Pyrogas is withdrawn by a fan, which maintains reactor's pressure slightly below atmospheric. The condensation unit consists of two in-series, water-cooled, surface heat exchangers, which enable condensation of the aqueous phase and organic fraction of the pyrogas. A demister is located upstream the fan to remove the remaining impurities in the pyrogas. A scheme of the reactor is reported in Figure 1.

The operating conditions of the slow pyrolysis test on pilot scale (Table 4) were set based on the laboratory scale results obtained during previous research activities on industrial sludge performed by RE-CORD ([29] and master thesis work: Di Bianca M., "Design of a thermo-chemical treatment plant for critical raw materials recovery from industrial sludge", University of Florence, 2021). Before being processed, the feedstock was dried at 105 °C. The solids residence time was set to 30 min. The operating temperature was set to 450 °C.

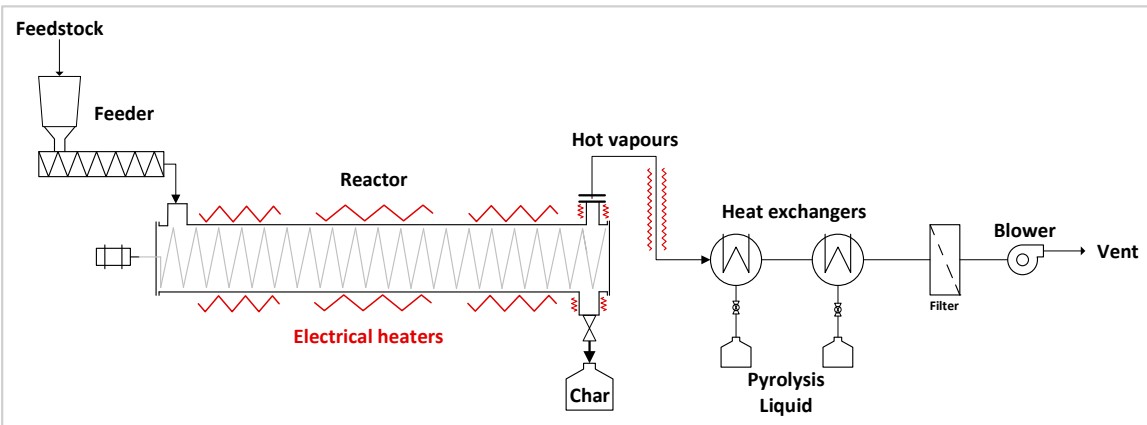

**Figure 1.** Simplified P&ID of the pilot scale pyrolysis reactor.

**Table 4.** Slow pyrolysis conditions adopted for the experiment.

| Parameter | Value | Unit |
|---|---|---|
| Temperature | 450 | °C |
| Residence time | 30 | min |

*2.4. Methodology of Chemical Leaching Experiments*

A chemical leaching experimental campaign was conducted at laboratory scale to extract and separate the desired inorganic elements from the pyrolyzed sludge. Selection of the chemical leaching process parameters was based on results from the literature [30], previous tests performed [29], and the mentioned thesis work and adapted accordingly. The aim was to maximize extraction of the most valuable elements (P, Al, K, Mg) from the feedstock in the resulting eluates and to obtain an upgraded char ("biocoal"). Different acids (nitric acid, $HNO_3$, and hydrochloric acid, HCl), and alkalis (potassium hydroxide, KOH) were used for the leaching tests. $HNO_3$ and HCl were dosed via a 64–66% $HNO_3$ solution and a 35–39% HCl solution, respectively. KOH was provided as pellets (assay 85–100). The following operating conditions were varied during the experiments:

- mass ratio between leaching solution and processed char (liquid:char);
- reagent concentration in the leaching solution, expressed as weight % $w/w$ (% reagent);
- operating temperature;
- contact time.

The mass ratio between the dosed reagent (acid/alkali) and the processed char (reagent:char) was used to compare the reagent dosage among different tests. For each test, the char from the slow pyrolysis pilot plant (char SPYRO), previously grinded, was added to the leaching solution prepared by mixing demineralized water and the selected acid or alkali in a beaker. The beaker was then covered on top and put on a heated plate equipped with a thermocouple and a magnetic stirrer for the set retention time. The mixture was then filtered by vacuum with a 1-micron paper filter. After the separation, the solid material was washed with demineralised water, newly separated from the liquid phase, and dried at 105 °C in an oven until constant weight. The two products of the process, i.e., the liquid and the solid phase, were then analysed. In stepwise tests, the leached char from first step was then subjected to the successive leaching step following the same methodology. The eluates obtained from the leaching processes of case 1 and case 3 were subjected to chemical precipitation. Known volumes of KOH or $CaCl_2$ solutions were added to the eluate under magnetic stirring while pH was monitored by a pH-meter (Metrohm 827 pH) during the process. In both cases, the precipitated solid was then separated from the liquid phase by centrifugation and finally dried in an oven at 105 °C.

In one case (case 1), KOH was added to the eluate to maximise precipitation of all inorganic compounds. The precipitation by KOH was stopped when a pH of 6–8 was reached in the liquid phase. In a different case (case 3), KOH was used as leaching reagent, while calcium chloride ($CaCl_2$) was added to the second eluate to cause the separation of the dissolved compounds of interest in solid form. $CaCl_2$ was provided as powder (assay $\geq$ 96). For the precipitation test by $CaCl_2$, the dosage of the reagent was calculated based on a set molar ratio between the calcium to be added and the phosphorous to be removed in the eluate. The improvement process carried out during the experimental campaign led to identification of the three processes (cases), including one (case 1) or more (cases 2, 3) chemical leaching step(s) followed by an eventual chemical precipitation phase (cases 1, 3). The processes involved in the cases, described in the following paragraphs, had a common aim: reduction of char mineral matter, with a separation of the most valuable inorganics, and the consequent increase of the biocoal carbon percentage content.

2.4.1. Case 1

In case 1 (Figures S1 and 2), a single-step chemical leaching process was performed, followed by a precipitation step of the eluate. The aim was to obtain high extraction efficiency of the inorganic compounds in solid form and to increase the carbon content of the char. The selected optimal operating conditions for the process were adapted to the studied material. In this case, $HNO_3$ was used as leaching agent, diluted in demineralized water at 6.3% $w/w$. Char was then added to the acid solution, with a liquid to char ratio of 10:1 and a retention time of 2 h. The mixture was maintained at 75 °C for 2 h. After the leaching test, to recover an inorganic compound of high quality for fertilizers production, a 15% KOH solution was used for the chemical precipitation step after the leaching test.

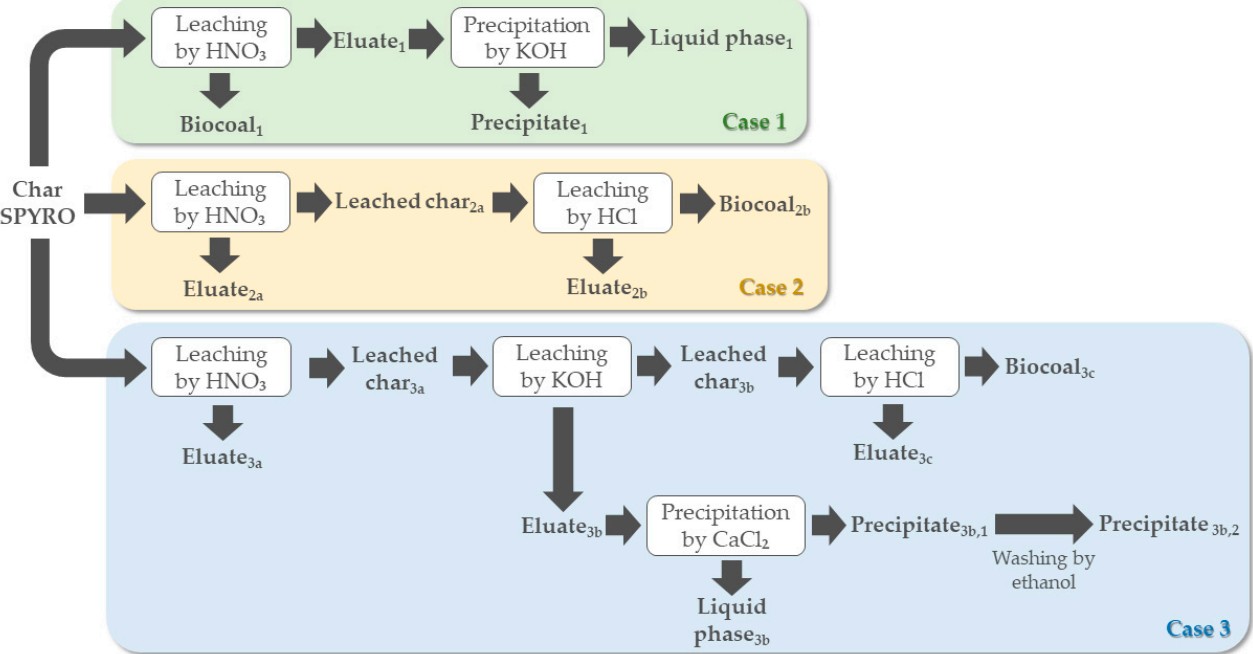

**Figure 2.** Case 1, case 2, and case 3 process schemes.

2.4.2. Case 2

Case 2 (Figures S2 and 2) consisted of a two-step acid leaching test. The aim was to obtain a final char with a lower inorganic content compared to the char of case 1 and verify the partition of P and Al among the eluates of the first and the second steps. In the second step, HCl was used as leaching agent to remove Al, as suggested by Valeev et al. [31]. A 0.5 M $HNO_3$ solution was used for the first step with a liquid to solid ratio of 20:1, with the

same acid to char ratio of case 1. For the second step, a 1 M solution of HCl was used with a liquid to solid ratio of 10:1. In both steps, the mixture was maintained at 75 °C for 2 h.

2.4.3. Case 3

Case 3 was developed to better separate Al and P to recover the two elements separately. The process was based on a process for selective recovery of phosphorous and aluminium from sewage sludge ash [30] and involved sequential use of an acid and alkali. The process described by [30] consists of acidic pre-treatment to dissolve the Ca present in the char, at mild conditions, to avoid removing P and Al; the second step consisted of alkali leaching. In fact, according to [30], the P and Al present as aluminium phosphate can be dissolved in alkaline conditions. Then, precipitation of the eluate from the alkali leaching step, performed by $CaCl_2$, enables separated recovery of P as calcium phosphate, leaving Al dissolved in the remaining liquid, with potential of application in wastewater treatment plants. In fact, while Ca should react with P to form calcium phosphate, Cl and Al remain in the solution and can be used to obtain aluminium polychloride [30].

The process reported by [30] was adapted, using different acid and alkali reagents, aiming to maximize recovery of products with a potential industrial application, increasing their value and minimizing the waste produced by the process. Case 3 included a first acidic pre-treatment by $HNO_3$, followed by an alkaline leaching step by KOH, and the selective precipitation by $CaCl_2$. In addition, a third leaching step with HCl included to reduce the biocoal ash content and recover the residual aluminium. The $HNO_3$ concentration in the solution for the pre-treatment was set to 0.2 M. Use of $HNO_3$ produced a N-rich solution and opens the possibility to apply precipitation on the eluate from the first step by KOH or $Ca(OH)_2$ in view of recovering the dissolved Ca as calcium nitrate, which could be applied for fertilizers production [32]. The same principle was followed in the second alkaline step, using KOH, with a temperature maintained at 60 °C. Although the expected result of this precipitation is the selective separation of P and Al, the difference is in the expected quality of the precipitate. Using KOH determines the presence of K in the precipitate, increasing the value of the recovered compound in the fertilizers sector since K is a key soil macronutrient. Chemical precipitation was applied on the KOH eluate by a 35% $CaCl_2$ solution and selecting a molar Ca:P ratio of 1.5. A part of the obtained precipitate was finally washed by ethanol in order to dissolve the residual impurities on the solid material. To guarantee significant ash reduction from the processed char, a third acidic HCl leaching step was included in the process, applying a 10:1 leaching solution to char ratio, a 1 M HCl concentration of the solution, 80 °C, and 2 h as operating conditions. Case 3 process scheme is reported in Figures S3 and 2 below, the latter showing case 1 and case 2 process schemes as well.

The leaching conditions adopted for the 3 cases are reported in Table 5.

**Table 5.** Operating conditions of the chemical leaching tests (cases 1–3) in the experimental campaign (char SPYRO: char from SPYRO pyrolysis pilot plant; leached char$_{2a}$: from case 2, step a; leached char$_{3a}$: from case 3, step a; leached char$_{3b}$: from case 3, step b).

| Case | Case 1 | Case 2 | | Case 3 | | |
|---|---|---|---|---|---|---|
| Leaching step | a | a | b | a | b | c |
| Starting material | Char SPYRO | Char SPYRO | Leached char$_{2a}$ | Char SPYRO | Leached char$_{3a}$ | Leached char$_{3b}$ |
| Reagent | $HNO_3$ | $HNO_3$ | HCl | $HNO_3$ | KOH | HCl |
| Liquid:char | 10:1 | 20:1 | 10:1 | 10:1 | 10:1 | 10:1 |
| % reagent | 6.3% $w/w$ (1 M) | 3.2% $w/w$ (0.5 M) | 3.7% $w/w$ (1 M) | 1.3% $w/w$ (0.2 M) | 4.0% $w/w$ (0.7 M) | 3.7% $w/w$ (1 M) |
| Reagent:char | 0.63 | 0.63 | 0.37 | 0.13 | 0.40 | 0.37 |
| Temperature | 75 °C | 75 °C | 75 °C | 75 °C | 60 °C | 80 °C |
| Contact time | 2 h | 2 h | 2 h | 2 h | 2 h | 2 h |

Precipitation tests were performed on two eluates produced: the eluate obtained by the single-step leaching of case 1 and the eluate obtained by the alkali leaching performed in case 3. The precipitation conditions are reported in Table 6.

**Table 6.** Operating conditions of the precipitation tests for case 1 and case 3 in the experimental campaign (eluate$_1$: from case 1, step a; eluate$_{3b}$: from case 3, step b).

| Case | Case 1 | Case 3 |
|---|---|---|
| Starting eluate | Eluate$_1$ | Eluate$_{3b}$ |
| Reagent | KOH | CaCl$_2$ |
| % reagent | 15% $w/w$ | 35% $w/w$ |
| Final pH | 6–8 | Not set |
| Molar Ca:P | Not set | 1.5 |
| Additional treatment | Not performed | Washing by ethanol |

*2.5. Extraction Tests Performances*

The performances of the leaching tests were evaluated by two indicators. The first is the ash extraction efficiency (*AE*), which expresses the percentage of extracted ash against the initial content in the processed char:

$$AE(\%) = \frac{ash\,710\,char,i\,(g) - ash\,710\,char,f\,(g)}{ash\,710\,char,i\,(g)} \cdot 100 \tag{1}$$

where *ash 710 char,i* is the ash 710 mass in the processed char and *ash 710 char,f* is the ash 710 mass of the produced char. This parameter was calculated starting from the ash 710 content (% d.b.) of the materials and their mass (g). The second indicator is the element extraction efficiency (*EE,l*), which expresses the percentage of a specific element in the eluate against the initial content of the element in the processed char:

$$EE,l(\%) = \frac{element\,eluate,f\,(g)}{element\,char,i\,(g)} \cdot 100 \tag{2}$$

where *element eluate,f* is the element mass extracted and dissolved in the eluate and *element char,i* is the element mass in the processed char. The *element mass eluate,f* was calculated multiplying the concentration of the element in the eluate (determined by MP-AES, in mg/kg) by the mass of the leaching solution, which was dosed for the process (kg). The *element mass char,i* was calculated starting from the concentration of the element in the char (in mg/kg) and the char mass processed in the first leaching step (kg).

For chemical precipitation tests, the element extraction efficiency (*EE,p*) was considered and calculated as follows:

$$EE,p(\%) = \frac{element\,precipitate,f\,(g)}{element\,eluate,i\,(g)} \cdot 100 \tag{3}$$

where *element eluate,i* is the element mass in the processed eluate (calculated starting from the processed mass of the eluate and the element concentration in the eluate determined analytically) and *element precipitate,f* is the element mass recovered in the precipitate. This value is determined considering the elemental composition and mass of the liquid phase produced by precipitation and calculated as the difference between the starting mass of the element and the mass left in the aqueous phase.

*2.6. Statistical Analysis*

Normality and homogeneity of parameters were tested prior to ANOVA, and data were normalized by transformation as needed. Data on biocoals characterization (i.e., *AE*, ash 710, carbon, hydrogen nitrogen content, molar H/C ratio) were processed with one-way

analysis of variance (ANOVA) followed by Tukey test at 95% confidence level (Minitab$^{\circledR}$ 17.1.0, Minitab Inc., State College, PA, USA).

## 3. Results and Discussion

### 3.1. Results of Slow Pyrolysis Tests

During the slow pyrolysis test on the pilot plant, 2.790 kg dry sludge was processed and 1.615 kg char (char SPYRO) produced, corresponding to a char yield of 57.9%. The elevated ash content in the feedstock (Table 2) leads to a higher char yield compared to the typical char yield of 20–35% usually achieved by slow pyrolysis [15]. The mass balance of the process is reported in Table 7, where output permanent gases mass is calculated as the difference between input dry sewage sludge mass and output char and liquids, which were weighted.

**Table 7.** Mass balance of the dry feedstock (sewage sludge) slow pyrolysis test on SPYRO pilot plant.

| Material | Mass (kg) | Percentage |
|---|---|---|
| Input dry sewage sludge | 2.790 | 100% |
| Output char | 1.615 | 57.9% |
| Output liquids | 0.640 | 22.9% |
| Output permanent gases | 0.535 | 0.2% |

The energy balance of the process (Table 8) shows the chemical power distribution of the feedstock among the pyrolysis products. The chemical power of the feedstock and char was calculated from the mass of the materials (respectively processed and produced, from Table 7) and the respective HHV (from Table 8). The chemical power of the liquids was calculated accordingly, referring to the mass of recovered liquids and liquids HHV calculated (from Tables 7 and 8, respectively). The chemical power of the output of permanent gases was determined as the difference between the input power and recovered power in the char and liquids. By 8.0 kWt theoretically introduced in the plant as dry sewage sludge, 5.5 kWt (almost 69%) is recovered as pyrogas (mainly as permanent gases) and 2.5 kWt is recovered as char.

**Table 8.** Energy balance of slow pyrolysis test on SPYRO pilot plant.

| Parameter | HHV (MJ/kg) | Chemical Power (kWt) | Percentage |
|---|---|---|---|
| Input dry sewage sludge | 10.3 | 8.0 | 100% |
| Output char | 5.6 | 2.5 | 31.5% |
| Output liquids | 3.7 | 0.7 | 8.2% |
| Output permanent gases | 32.3 | 4.8 | 60.2% |

### 3.2. Characterization of the Slow Pyrolysis Char

The physical–chemical analysis of the char obtained (Table 9) shows an increase in ash content of about 60% against the processed feedstock. During pyrolysis, part of the organic matter is devolatilised and converted into pyrogas, determining the increase in the inorganic compounds (ash) percentage content [33] and leading volatiles to decrease to 12.5%. The material has a carbon content of about 14.4% and a nitrogen content of 2.4%. Char LHV is reduced compared to the dry feedstock due to the increased ash content and decrease in carbon and hydrogen in the material.

The elemental composition of the char in terms of metals and other mineral elements is reported in Table 10. The sum of Al (14.1%), Si (10.0%), P (47.2%), Ca (3.6%), and Fe (2.6%) represents around 35% of the char dry matter. Comparing this composition with the composition of sewage sludge (Table 3), we can observe an increase in almost all concentrations. In particular, aluminium concentration is almost tripled, and chromium

and copper concentrations show increases of around 80% and 60%, respectively. The iron concentration is almost doubled, while nickel and lead are almost two and three times, respectively, compared to sewage sludge.

**Table 9.** Results of the char (from SPYRO pilot plant) proximate, ultimate, and thermal analysis.

| Parameter | Value | Unit |
|---|---|---|
| Moisture | 1.2 | % w.b. |
| Ash 550 | 79.0 | % d.b. |
| Ash 710 | 77.7 | % d.b. |
| Volatiles | 12.5 | % d.b. |
| Fixed C | 9.9 | % d.b. |
| C | 14.4 | % d.b. |
| H | 0.9 | % d.b. |
| N | 2.4 | % d.b. |
| S | 0.02 | % d.b. |
| Cl | 0.05 | % d.b. |
| Molar H/C | 0.8 | - |
| HHV | 5.6 | MJ/kg d.b. |
| LHV | 5.4 | MJ/kg d.b. |

Notes: w.b.—wet basis. d.b.—dry basis.

**Table 10.** Elemental composition of char (from SPYRO pilot plant).

| Element | Value | Unit |
|---|---|---|
| Al | 142,848 | mg/kg d.b. |
| B | 7 | mg/kg d.b. |
| Ba | 1137 | mg/kg d.b. |
| Ca | 36,598 | mg/kg d.b. |
| Cr | 118 | mg/kg d.b. |
| Cu | 1122 | mg/kg d.b. |
| Fe | 26,227 | mg/kg d.b. |
| K | 8151 | mg/kg d.b. |
| Li | 28 | mg/kg d.b. |
| Mg | 8359 | mg/kg d.b. |
| Mn | 601 | mg/kg d.b. |
| Na | 1317 | mg/kg d.b. |
| Ni | 156 | mg/kg d.b. |
| P | 47,732 | mg/kg d.b. |
| Pb | 112 | mg/kg d.b. |
| Si | 100,318 | mg/kg d.b. |
| Ti | 261 | mg/kg d.b. |
| V | 43 | mg/kg d.b. |
| Zn | 704 | mg/kg d.b. |

Note: d.b.—dry basis.

### 3.3. Characterization of the Slow Pyrolysis Liquids

Characterization of the liquids obtained from the slow pyrolysis test is reported in Table 11. The elevated water content (almost 86%) affects the heating value of the material, which is very low.

**Table 11.** Results of liquids (from SPYRO pilot plant) analysis.

| Parameter | Value | Unit |
|---|---|---|
| Water content | 85.6 | % w.b. |
| C | 7.9 | % w.b. |
| H | 10.7 | % w.b. |
| N | 0.9 | % w.b. |
| HHV | 3.7 | MJ/kg w.b. |
| LHV | 1.4 | MJ/kg w.b. |

Note: w.b.—wet basis.

### 3.4. Tests of Element Extraction and Inorganic Products Characterization

In this paragraph, the evaluation of the extraction performances of the 3 cases is reported. The characterization of the obtained products is reported as well. The ash extraction efficiency (*AE*) and element extraction efficiency (*EE,l* and *EE,p*) of the processes involved in the 3 cases are summarized in Figure 3.

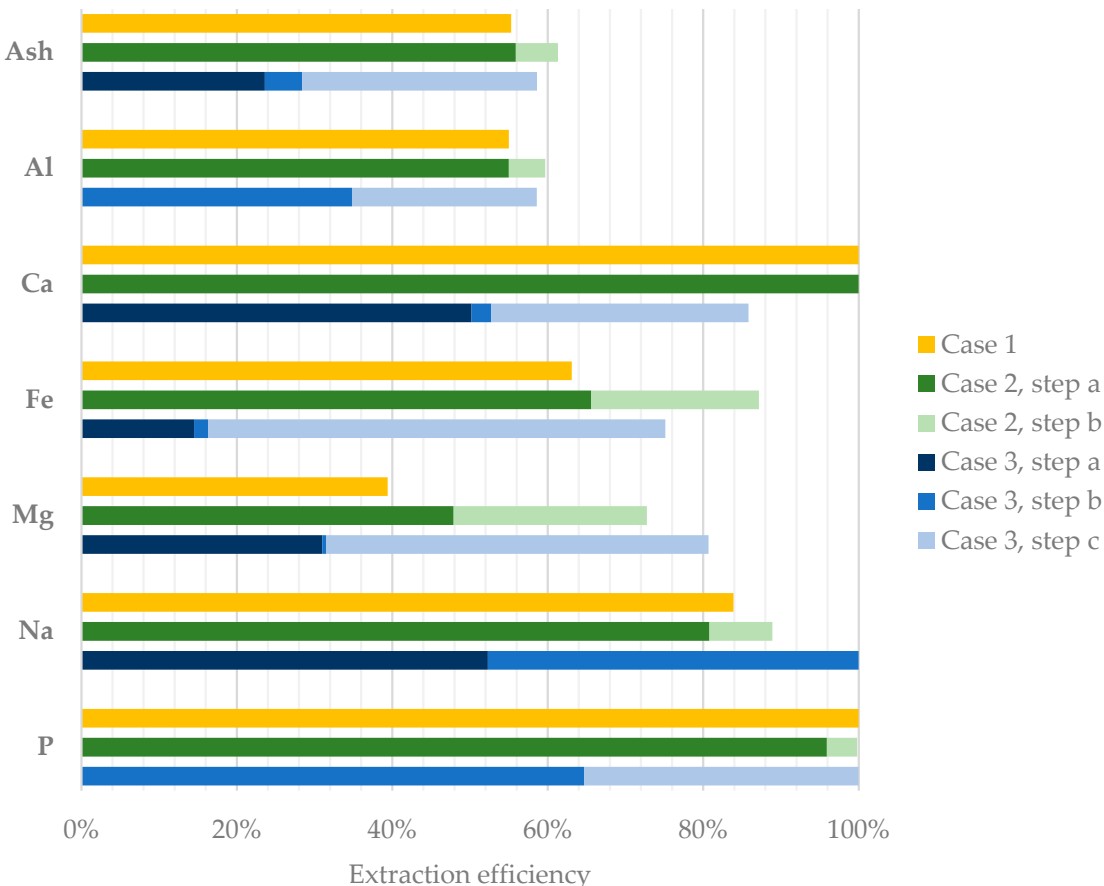

**Figure 3.** Extraction efficiency of ash and main inorganic elements achieved by the performed test cases (1–3).

### 3.4.1. Case 1

Figure 3 shows that 100% of Ca and P were extracted from the char, and around 63% of Fe and 84% of Na were extracted as well. Further, 55% of Al was extracted, and extraction efficiencies under 40% were achieved for K and Mg. Silicon remained in the char.

Chemical precipitation by KOH solution led to recovery of 100% of Al and P, around 47% of Ca, and about 92% of Fe from the eluate from the leaching step (Table 12). To bring pH of the eluate from 1.8 to 6.1 (Table 12), 0.07 g KOH/g eluate was dosed. The produced precipitate is rich in K (13.1%), P (5.4%), and N (4.7%), which are macronutrients for plants, and Ca (1.7%) and Fe (1.5%), which can also be applied for plant fertilization. However, Al is also present in the precipitate (9.1%).

**Table 12.** Characterization of case 1 eluate (eluate$_1$) and precipitated inorganic product (precipitate$_1$).

| Parameter | Eluate$_1$ | Precipitate$_1$ | Unit |
|---|---|---|---|
| pH | 1.8 | n.d. | - |
| N | n.d. | 4.7 | % |
| Al | 7787 | 90,794 | mg/kg |
| B | 3 | b.d.l. | mg/kg |
| Ba | 12 | 115 | mg/kg |
| Ca | 3025 | 17,048 | mg/kg |
| Cr | 5 | 37 | mg/kg |
| Cu | 1 | 37 | mg/kg |
| Fe | 1761 | 15,065 | mg/kg |
| K | 261 | 130,739 | mg/kg |
| Li | 2 | 11 | mg/kg |
| Mg | 336 | 3026 | mg/kg |
| Mn | 66 | 407 | mg/kg |
| Na | 116 | 1150 | mg/kg |
| Ni | 6 | b.d.l. | mg/kg |
| P | 5866 | 54,085 | mg/kg |
| Pb | 5 | b.d.l. | mg/kg |
| Si | 457 | 3940 | mg/kg |
| Ti | 3 | 42 | mg/kg |
| V | 1 | 8 | mg/kg |
| Zn | 63 | 560 | mg/kg |

Notes: b.d.l.—below detection limit. n.d.—not determined.

The benefits of this process are complete P and Ca removal from the char and good char ash extraction (around 55%), which can be reached by an operatively simple process. In addition, eluate Al, P, and Ca are totally recovered in solid form. The negative aspects of this acid leaching process are related to simultaneous extraction of Ca, P, and Al. In fact, the presence of Al in the produced precipitate is undesired when the precipitate is applied to produce P-rich fertilizers. Despite aluminium not being harmful, nor avoided in agriculture, its presence in high concentration reduces the quality and market value of the precipitate as a precursor for inorganic fertilizers production.

### 3.4.2. Case 2

By $HNO_3$ leaching, 100% Ca and almost 96% P were extracted, but 55% Al as well, with the same performances of case 1. During step b, 4.7% Al was extracted, and the process was not effective on Si extraction. As shown in Table 13, the eluate from $HNO_3$

leaching contains 0.2% P, 0.2% Ca, and 0.4% Al, which are the main elements. Precipitation of inorganic compounds from the eluate by $HNO_3$ leaching was not performed as it was considered equivalent to the one of case 1. The eluate from HCl leaching contains 0.1% Al as the main compound. The foreseen process for precipitation of the eluate from HCl leaching, although not tested, was a NaOH chemical precipitation to recover Al in the form of aluminium polychloride. No effective separation between Al and P in the two eluates resulted from this test. However, the second step with HCl contributed to increase the ash extraction efficiency from the input char, achieving an *AE* of 61.4%.

**Table 13.** Characterization of case 2 eluate from step a (eluate$_{2a}$) and step b (eluate$_{2b}$).

| Parameter | Eluate$_{2a}$ | Eluate$_{2b}$ | Unit |
|---|---|---|---|
| pH | $\leq 2$ | $\leq 2$ | - |
| Al | 3877 | 1166 | mg/kg |
| B | 1 | 0.2 | mg/kg |
| Ba | 5 | 24 | mg/kg |
| Ca | 2008 | 98 | mg/kg |
| Cr | 2 | 5 | mg/kg |
| Cu | 1 | 81 | mg/kg |
| Fe | 850 | 987 | mg/kg |
| K | 132 | 138 | mg/kg |
| Li | 1 | 1 | mg/kg |
| Mg | 198 | 363 | mg/kg |
| Mn | 29 | 7 | mg/kg |
| Na | 53 | 19 | mg/kg |
| Ni | 3 | 7 | mg/kg |
| P | 2260 | 328 | mg/kg |
| Pb | 2 | 6 | mg/kg |
| Si | 302 | 26 | mg/kg |
| Ti | 1 | 6 | mg/kg |
| V | 1 | 3 | mg/kg |
| Zn | 33 | 10 | mg/kg |

### 3.4.3. Case 3

By mild $HNO_3$ leaching, 50.2% Ca and 52.3% Na were dissolved into the eluate (Figure 3), while 100% Al and P were left in the char. KOH leaching led to extraction of 34.8% Al and 64.7% P, which were the targets of the process. In fact, KOH eluate consisted mainly of K (17.6%), Al (1.0%), and P (0.4%) (Table 14). In the precipitation phase (Figure S4), 0.02 g $CaCl_2$/g eluate was dosed to have a Ca:P molar ratio of 1.5, optimal for calcium phosphate recovery, as suggested by the literature [30,34,35]. The elemental compositions of the precipitate (precipitate$_{3b,1}$ in Table 14) shows that the material contains a relevant quantity of Ca (13.6%), K (14.0%), and P (2.7%) but has an important Al content as well (4.9%). A minor concentration of Na (0.5%) is present. Since 43.7% of Al was recovered in the solid precipitate (Figure 3), we can conclude that around 56% of Al remained in the liquid phase.

**Table 14.** Characterization of case 3 inorganic products from step a (eluate$_{3a}$), step b (eluate$_{3b}$, precipitate$_{3b,1}$, precipitate$_{3b,2}$, liquid phase$_{3b}$), step c (eluate$_{3c}$).

| Parameter | Eluate$_{3a}$ | Eluate$_{3b}$ | Precipitate$_{3b,1}$ | Precipitate$_{3b,2}$ | Liquid Phase$_{3b}$ | Eluate$_{3c}$ | Unit |
|---|---|---|---|---|---|---|---|
| Al | b.d.l. | 10,254 | 48,998 | b.d.l. | 6622 | 9860 | mg/kg |
| B | 1 | b.d.l. | b.d.l. | b.d.l. | b.d.l. | b.d.l. | mg/kg |
| Ba | 1 | 7 | 63 | 162 | 6 | 56 | mg/kg |
| Ca | 1813 | 113 | 135,506 | 121,003 | 122 | 1536 | mg/kg |
| Cr | b.d.l. | 0.4 | b.d.l. | b.d.l. | b.d.l. | 8 | mg/kg |
| Cu | b.d.l. | b.d.l. | b.d.l. | b.d.l. | 1 | 27 | mg/kg |
| Fe | 376 | 58 | 337 | 782 | 56 | 1954 | mg/kg |
| K | 147 | 175,767 | 139,923 | 348 | 16,113 | 5157 | mg/kg |
| Li | 2 | b.d.l. | 1 | b.d.l. | b.d.l. | 1 | mg/kg |
| Mg | 256 | 5 | 52 | b.d.l. | 5 | 521 | mg/kg |
| Mn | 25 | 1 | b.d.l. | b.d.l. | 1 | 45 | mg/kg |
| Na | 68 | 193 | 5388 | 6 | 166 | 96 | mg/kg |
| Ni | b.d.l. | b.d.l. | b.d.l. | b.d.l. | b.d.l. | 11 | mg/kg |
| P | b.d.l. | 3734 | 26,878 | 73,212 | 2974 | 2870 | mg/kg |
| Pb | b.d.l. | b.d.l. | b.d.l. | b.d.l. | 1 | 10 | mg/kg |
| Si | 54 | 61 | 630 | 1524 | 58 | 450 | mg/kg |
| Ti | b.d.l. | b.d.l. | b.d.l. | b.d.l. | b.d.l. | 4 | mg/kg |
| V | b.d.l. | 1 | b.d.l. | b.d.l. | 2 | 2 | mg/kg |
| Zn | 4 | 2 | b.d.l. | 29 | 2 | 55 | mg/kg |

Notes: b.d.l.—below detection limit.

Comparing the composition of this precipitate (precipitate$_{3b,1}$) and the ethanol-washed precipitate (precipitate$_{3b,2}$ in Table 14), we can observe that, by precipitate washing by ethanol, a reduction in the concentration of some elements has been achieved. In particular, almost the total amount of Al was removed, the K concentration was reduced to 0.03%, and Ca was reduced to 12.1%. This effect suggests that these elements could not be chemically bonded in the solid precipitate and that a material of high P and Ca content and a lower level of impurities can be obtained. On the other hand, the extraction efficiencies of P and Ca achieved by precipitation (30.6% and 10.9%, respectively) are quite modest contrary to the target and in comparison to case 1, concluding that the prevalent part of Ca and P was left in the separated liquid phase.

The third leaching step by HCl was effective on the remaining Al and P, which were totally extracted, and partly on Ca, whose $EE,l$ was around 33%. The produced HCl eluate (Figure S5) contains 1.0% Al, 0.5% K, 0.3% P, 0.2% Ca, and 0.2% Fe (Table 14).

### 3.5. Ash Extraction Efficiency and Biocoals Characterization

The ash extraction efficiencies ($AE$) achieved in cases 1–3 are summarized in Table 15, showing that the mean values were statistically different. We can observe that the highest $AE$ (61.4%) was achieved by the sequential HNO$_3$ and HCl leaching of case 2. The first step of this case and the single-step HNO$_3$ leaching of case 1 are comparable (around 55%) since the only different adopted parameter was the liquid to char ratio, which did not have a high impact on the process. Case 3, which included sequential leaching by HNO$_3$, KOH, and HCl, led to an overall $AE$ around 58.6%. The third step by HCl, in particular, was the most effective on ash extraction among the three of the case, and its introduction in the process was dictated by the low $AE$ obtained by the first two steps.

**Table 15.** Ash extraction efficiency (*AE*) and characterization of the extracted biocoals case 1 (biocoal$_1$), case 2 (biocoal$_{2b}$), and case 3 (biocoal$_{3c}$). Different letters in the same row indicate significant differences between means for $p < 0.05$ ($n = 3$).

| Parameter | Biocoal$_1$ | Biocoal$_{2b}$ | Biocoal$_{3c}$ | Unit |
|---|---|---|---|---|
| *AE* | 55.3 c | 61.4 a | 58.6 b | % d.b. |
| Ash 710 | 60.9 a | 57.4 c | 59.0 b | % d.b. |
| C | 25.5 c | 27.4 a | 26.7 b | % d.b. |
| H | 1.5 b | 1.7 a | 1.4 a | % d.b. |
| N | 4.2 a | 4.1 a | 4.1 b | % d.b. |
| Molar H/C | 0.7 c | 0.7 b | 0.6 a | % d.b. |
| Surface area * | 37 | 70 | 42 | m$^2$/g |

Notes: d.b.—dry basis. * for this parameter, the data refer to one replicate only.

In Table 15, the characterizations of the produced biocoals are provided as well, while Figure 4 shows the biocoal from case 3. Since the ash content of the processed char was around 77.7% (Table 9), although the achieved *AE* was over 55% for case 1, case 2 and case 3, the ash content of the biocoals produced remained over 50%. Case 2 led to the minimum ash content, 57.4%. The ash mean value of biocoal$_1$ was significantly higher than that of biocoal$_{3c}$ and biocoal$_{2b}$, the latter showing the lowest value.

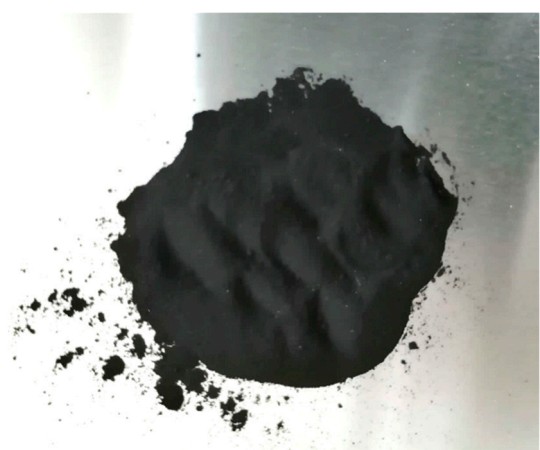

**Figure 4.** Biocoal from step c of case 3 (biocoal$_{3c}$).

Ash extraction from the processed char by leaching led to increasing carbon content in the material since one of the targets of the integration of slow pyrolysis and chemical leaching was to recover carbon from sewage sludge. Table 15 shows that the carbon percentage content of the biocoals from case 1, case 2, and case 3 was increased, respectively, by 77%, 91%, and 86% against the starting carbon content in the processed char. We can observe that higher carbon content was reached when a higher *AE* was achieved, so case 2, case 3, and case 1 led to statistically different carbon contents of 27.4%, 26.7%, and 25.5%, respectively. Similarly, hydrogen and nitrogen contents of biocoals are quite comparable (respectively, between 1.4% and 1.7% and between 4.1% and 4.2%). Statistical analysis showed that the mean values of hydrogen content of biocoal$_{2b}$ and biocoal$_{3b}$ were similar but higher than case 1 biocoal. On the other hand, biocoal$_{3b}$ showed the lowest nitrogen content, the values of biocoal$_{2b}$ and biocoal$_1$ being significantly similar. The molar H/C ratio is around 0.6 (case 3) and 0.7 (cases 1 and 2), the values being statistically different.

The mineral composition of the biocoal obtained by EDX analysis is reported in Table 16. We can observe that SiO$_2$ is the main compound, representing 29–42% of the biocoals, followed by minor concentrations of Al$_2$O$_3$, representing above 10–15% of the biocoals. In fact, none of the leaching tests were effective on Si extraction from char.

**Table 16.** Ash composition of biocoals (from cases 1–3).

| Oxide | Case 1 | Case 2 | Case 3 | Unit |
|---|---|---|---|---|
| $SiO_2$ | 67 | 61 | 71 | % ash |
| $Al_2O_3$ | 16 | 19 | 19 | % ash |
| $Fe_2O_3$ | 3 | 6 | 4 | % ash |
| $K_2O$ | 3 | 3 | 4 | % ash |
| Other | 11 | 11 | 3 | % ash |

Concerning case 2, an interesting surface area value, namely 70 m$^2$/g, was reached, which is higher than typical waste material values and comparable to that shown by biochar produced from some lignocellulosic materials (e.g., maize straw [36]). This characteristic makes the material of potential interest for being used as absorption material or activated carbon production.

The char produced from case 3 presented reduced surface area compared to case 2 (42 m$^2$/g), probably due to a higher ash content. As visible from Table 15, surface area increases with a decrease in ash content in the biocoal [37]. The pore size distribution of the biocoals, determined by DFT method, is reported in Figure 5, showing an increase in the cumulative pore volume (analysed in the range 0–50.2316 nm) from biocoal$_1$ to biocoal$_{2b}$, similar to surface area.

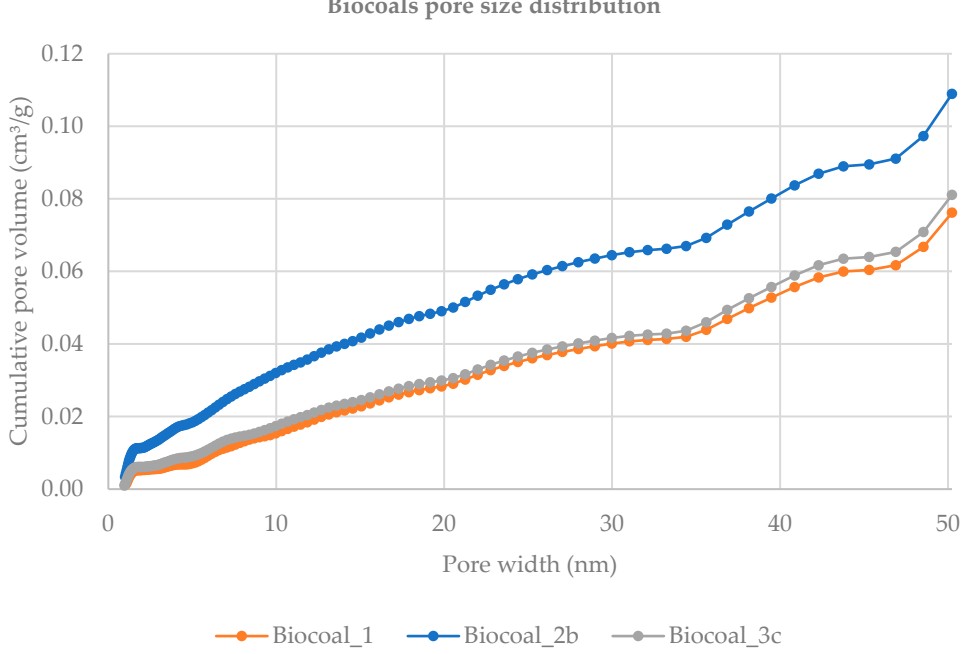

**Figure 5.** Pore size distribution of the biocoals (cases 1–3) determined by DFT method.

As reported in Table 16, the ash of the extracted biocoals in the three cases contains mainly silica (from 61% to 71% of dry ash) and alumina (from 16% to 19% of dry ash), with the rest including $Fe_2O_3$ (3–6%) and $K_2O$ (3–4%), plus a variable amount of other inorganic compounds, such as carbonates, and other metal oxides.

As visible from Figure 3, which reports the performance in terms of ash and inorganic elements extraction rate obtained by the three performed test cases, high extraction efficiency was achieved for P and Ca in all the three cases, while the extraction efficiency of aluminium did not overcome 60%. Potassium extraction efficiency was not reported in Figure 3 as this parameter could be determined only for case 1 and case 2. For these two cases, the extraction efficiency of potassium was found to be 33.6% and 42.5%, respectively.

### 3.6. Discussion on Leaching Tests Performances

The first one-step chemical leaching test enabled to extract 100% of P and Ca and more than 50% of Al and Fe contained in char, with 55.3% of initial ash dissolved into eluate. Precipitation by KOH enabled to recover 100% Al and P in a promising solid N–P–K inorganic fertiliser. Due to the presence of aluminium, which reduces the quality of a recovered inorganic compound, two additional tests were performed with two and three leaching steps, respectively. In fact, although aluminium is not considered harmful for soil, its presence in an inorganic compound lowers concentration of other useful elements for soil (NPK), leading the quality of the fertilizing product to decrease. However, despite slightly higher ash extraction efficiency, double-step acid leaching by $HNO_3$ followed by HCl of case 2 enabled to extract around 55% Al, 100% Ca, and around 96% P without separate recovery of aluminium. In comparison with similar processes for combined recovery of carbon and inorganic elements, the process tested in this work has shown higher phosphorus extraction performances. In fact, a phosphorus extraction efficiency of 71% was achieved by acid leaching with HCl and subsequent precipitation with NaOH of hydrochar obtained from sewage sludge HTC [38], while use of $HNO_3$ as a leaching agent led to recovery of 78% of the phosphorous retained in the hydrochar from an Italian sewage sludge [39]. Despite a high phosphorus extraction rate, separation of Al from P was partially achieved in case 3 but with a complicated, ineffective process.

Analysing case 3, a leaching step by $HNO_3$ aimed at production of calcium nitrate. In the following step (leaching by KOH), the targets of extraction were P and Al in the form of aluminium phosphate, and around 35% Al and 65% P were extracted. The solid product obtained from precipitation by $CaCl_2$ applied to KOH eluate, washed with ethanol, had relevant Ca (7.3%) and P (12.1%) contents, which makes it potentially applicable as raw material for fertilizers production. However, phosphorus recovered in the precipitate was reduced compared to the total available as 30.6% of the P available in the eluate was recovered in the precipitate. Addition of a third leaching step by HCl enabled to reach 86% Ca and 100% P extraction from char, reducing the ash content of the generated biocoal. However, the third step performed with HCl produced a third eluate rich in chlorine, where most of the phosphorus and aluminium were dissolved, but of difficult valorisation in agriculture.

### 3.7. Potential Application of Obtained Biocoals

The quality of the biocoals produced by the three cases is comparable. All obtained biocoals presented a C content between 25 and 28% on a dry basis and an ash content between 57.4% and 61%. With ashes composed mainly of $Al_2O_3$ and $SiO_2$, the biocoal from case 2 presented an interesting BET surface area (70 $m^2/g$). The study demonstrated that, at the proposed conditions, maximum extraction of P and Ca is achievable, but silicon is not removed. Silica extraction by alkali leaching could be performed but at more severe conditions. For example, on coal fly ash, an extraction efficiency of $SiO_2$ around 80% was achieved at NaOH concentration of 40%, ratio of fly ash to NaOH 1:1, leaching temperature of 95 °C, and leaching time of one hour [40]. However, an increase in operating conditions severity was avoided as it could affect the organic matter of chars and could also impact the process economy.

A more interesting strategy could be represented by application of biocoal as precursor for adsorbents production, a $SiO_2$ source in the construction industry, or in metallurgy to produce silicon metal in submerged arc furnaces. In the case of silicon metal production, during the process, $SiO_2$ is reduced into silicon at high temperatures (>2000 °C). A reduction reaction takes place by adding a reducing agent to the silica. The reducing agent consists of carbon in the form of mineral carbon, or charcoal or wood chips [41]. Quartz sand and carbon are fed in appropriate proportions through the top, and liquid silicon is extracted at the bottom [42]. According to the simplified reaction of the process [41], 0.4 kg C is necessary per kg of $SiO_2$ reduced. The ratios between C and $SiO_2$ contained in the biocoals from case 1, case 2, and case 3 correspond to 0.6, 0.8, and 0.6, respectively, meaning that

they all contain an excess of C compared to the stoichiometric quantity. According to the reaction, C excess can also result in production of silicon carbide SiC [41].

## 4. Conclusions

In this work, integration of slow pyrolysis and chemical leaching for recovery of valuable resources (carbon, phosphorous, calcium, aluminium, and silicon metal) from sewage sludge was studied, aiming to obtain fungible products. The study demonstrated that 100% for Ca and 100% P could be extracted from raw char and that precipitation by KOH enabled to obtain a promising solid NPK inorganic fertiliser, which a fertiliser company has already validated as a valuable product.

A second result is that, as shown by experiments, with both acid and alkali leaching, at atmospheric pressure and temperatures below 80 °C, silica is not removed. Case 2 and case 3 demonstrated that, with more severe leaching conditions, higher ash extraction efficiency is achieved (61.4%) but without removing silica. Chemical extraction of silica could be performed but with probably too high equipment and operational costs.

Considering the similar composition of the biocoals obtained by the three cases and the reduced efficiency in terms of Al-P separation obtained by cases 2 and 3, adoption of single-stage leaching is considered more competitive for industrial application.

A potential solution to avoid the presence of aluminium in precipitated salt is to avoid use of aluminium polychloride in wastewater treatment. Replacing aluminium polychloride with iron chloride could facilitate separation during leaching. Moreover, iron is potentially useful for fertiliser production industries. Replacing Al-chloride with Fe-chloride during wastewater treatment could also provide relevant benefits to valorisation of biocoal.

Regarding the obtained biocoal, silica removal by flotation or mechanical or gravimetric separation could be tested in the future to reduce ash content. Alternatively, it is suggested to directly use biocoal as a combined source of C and $SiO_2$. Once phosphorus and other harmful elements (Na, S) are removed and valorised separately, the biocoal can be used in a set of industrial sectors, in particular the cement industry and silicon production industry. To this end, processing a sludge with lower Al content and performing more severe leaching could improve the quality of the biocoal as almost only $SiO_2$ could be remaining in the biocoal ash. Achievement of the required purity will enable to fully recover the silicon contained in the sludge and replace fossil coal used in silicon metal production with a renewable bio-based carbon source. The outcomes of this study are considered relevant both for scientists engaged in the study of sewage sludge pyrolysis and for industrial actors in the wastewater treatment sector as producers of sewage sludge and engaged in identification of sustainable valorisation technologies.

As an additional research pathway, use in the cement industry is considered a promising alternative. Furthermore, achieved surface area is considered a promising aspect for valorisation of a solid as an adsorbent for wastewater treatment itself. Therefore, further experiments on macro-porosity and biocoals adsorption capacity should be performed.

**Supplementary Materials:** The following supporting information can be downloaded at: https://www. mdpi.com/article/10.3390/w15061060/s1, Figure S1: Case 1 process scheme. Figure S2: Case 2 process scheme. Figure S3: Case 3 process scheme. Figure S4: Precipitation by $CaCl_2$ applied on the eluate from step b of case 3 (eluate$_{3b}$). Figure S5: Eluate from step c of case 3 (eluate$_{3c}$).

**Author Contributions:** All authors contributed to the study conception and design. Material preparation, tests conduction, data collection, and analysis were performed by A.S., M.D.B., and G.L. The first draft of the manuscript was written by A.S. and M.D.B. and edited and commented by G.L., A.M.R., and D.C. All authors have read and agreed to the published version of the manuscript.

**Funding:** Funding was provided by: POR FESR TOSCANA 2014–2020, AZIONE 1.1.5 sub A1)–FSC 2014–2020 (ex Del.CIPE 40/2020) Project "Aiuti agli investimenti R&S delle imprese" CUP: 3647.04032020.157000040; and H2STEEL–"Green H2 and circular bio-coal from biowaste for cost-

competitive sustainable Steel", funded by the European Commission HORIZON-EIC-2021-PATHFIN DERCHALLENGES-01-04-Novel routes to green hydrogen production.

**Data Availability Statement:** The most significant data generated and/or analysed during this study are included in this published article. Additional datasets are available from the corresponding author on reasonable request.

**Acknowledgments:** The authors acknowledge Lorenzo Iachini for his contribution to the experimental activity.

**Conflicts of Interest:** The authors declare no conflict of interest.

## Nomenclature

| | |
|---|---|
| Sewage sludge | the processed feedstock object of this study; |
| Char | the raw solid product obtained from sewage sludge slow pyrolysis; |
| Eluate | the liquid obtained after every char leaching step; |
| Precipitate | the inorganic salt obtained by chemical precipitation of inorganic compounds from eluate; |
| Leached char | the intermediate solid material, obtained by a leaching step, to be further processed as input material in a subsequent leaching step; |
| Biocoal | the solid final product obtained after upgrading of char by chemical leaching process |

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
