# Peer review of "Opportunities of Integrating Slow Pyrolysis and Chemical Leaching for Extraction of Critical Raw Materials from Sewage Sludge"

_water, doi:10.3390/w15061060_

Round 1

Reviewer 1 Report

The topic is interesting and current with a lot of presented results, however, it seems disjointed and lacks experiments and analyses.

Why did you use slow pyrolysis?

You wrote that sewage sludge must be dried up to 30% in order to go to pyrolysis, why? what happens if it is less or more than that? does it affect the yield or does some other chemical reaction occur that you did not expect or that affects the final product?

Why didn't you use hydrothermal carbonization (HTC) to save money?

You used chemical leaching to extract phosphorus, could you have used another method to extract other elements?

Paragraph 1.4

Row 136 porous solid with carbon? what did you mean, is it active carbon material

Why didn't you chemically wash the sewage sludge and then treat the dry residue with slow pyrolysis?

Agree terms, char, biocoal or maybe biochar?

In the experimental part, you mention a calorimetric bomb for pre-treatment and then for determining the calorimetric power and LHV, why didn't you do HHV?

You wrote that you did EDX in the experimental part, which usually goes with SEM, but I didn't see the results,

You presented BET results and did not provide isotherms or pore distribution

Why didn't you do FTIR of the solid residue to see the chemistry of the surface and functional groups of the solid residue?

You could do ICP to see the content of heavy metals

Author Response

The authors gratefully acknowledge the anonymous reviewer for its report, which offers the opportunity to better clarify on the aim and scope of the paper and the methodology adopted. Answers to all the reviewer’s questions and comments have been addressed, and reported below.

The topic is interesting and current with a lot of presented results, however, it seems disjointed and lacks experiments and analyses.

  • Why did you use slow pyrolysis?

Slow pyrolysis has been adopted as it represents the thermochemical conversion process which, in comparison with intermediate, fast and flash pyrolysis, maximizes the solid yield.. However, char obtained from slow pyrolysis presents higher hydrophobicity, higher stability, with lower oxygen content and, consequently, lower volatile content. These properties make the slow pyrolysis char similar to an anthracite, or to a fossil coal.

  • You wrote that sewage sludge must be dried up to 30% in order to go to pyrolysis, why? what happens if it is less or more than that? does it affect the yield or does some other chemical reaction occur that you did not expect or that affects the final product?

During pyrolysis, before the beginning of devolatilization reaction, water content of the feedstock is evaporated. High moisture content in the sludge has two damaging effects on pyrolysis process:

  • It increases the residence time needed to achieve the required char stability, because longer time is needed to evaporate water;
  • It brings to the production of pyrogas with high water content, thus with lower calorific value.
  • Why didn't you use hydrothermal carbonization (HTC) to save money?

It is recognized that HTC can reduce the energy consumption for drying. This technology, applicated to sewage sludge, has been tested and studied by the authors in previous projects[1],2. However, hydrochar is characterized by high volatiles content (>50 % db). The volatility of the hydrochar brings to oxydation reaction during leaching process. In fact, oxydizing reagents such as HNO3 react with the hydrochar organic matter, producing CO2 emissions and bringing organic matter in the leaching liquid, complicating the recovery of inorganic elements, Moreover,the study described in the article was developed in the context of a project where the industrial partner interested in the technology, a wastewater treatment company, has already a drying system. Thus, HTC was not in the scope of the project.

  • You used chemical leaching to extract phosphorus, could you have used another method to extract other elements?

Most of inorganic elements (Mg, K, Fe, Al) can be extracted by chemical leaching. However, as shown in the article, other elements such as silicates, and part of alumina, could not be extracted. In order to improve the ash reduction performances of the integrated process, alternative mechanical-gravimetric systems for extraction of silicates, such as flotation, could be tested. For this reason, authors are currently investigating these options, which could be included as part of a future manuscript.

Paragraph 1.4

  • Row 136 porous solid with carbon? what did you mean, is it active carbon material

With the definition “porous solid with carbon”, the authors wanted to describe the generic characteristics of the biochar produced by slow pyrolysis. According to the authors experience, the slow pyrolysis biochar, even if not activated, consists in a carbonaceous solid, with a porous structure, relevant surface area and microporosity. To increase the surface are and the pores volume, the char can be furtherly processed by chemical, or physical activation. However, char activation has not been tested in the proposed work

  • Why didn't you chemically wash the sewage sludge and then treat the dry residue with slow pyrolysis?

Similarly to what was discussed previously in the context of leaching applied to biochar from pyrolysis, due to the presence of volatile organic compounds in the raw sludge, processing it with an acid leaching brings to strong oxidation reactions. By consequence, organic matter reacts by producing both CO2, and organic compounds remain in the liquid. This brings to two detrimental effects: production of of gases to be treated, reduce effectiveness of leaching process (part of acid reacts with organic matter, and not with inorganic elements), bad quality of leached liquid, with organic matter dissolved.

  • Agree terms, char, biocoal or maybe biochar?

As biochar is used for agriculture, the use of this term has been avoided. Biocoal and char refer to two different materials: char is the raw solid produced after pyrolysis of the sludge, biocoal is the solid obtained after char leaching.

A disambiguation paragraph is provided at the beginning of the article to guide the reader.

  • In the experimental part, you mention a calorimetric bomb for pre-treatment and then for determining the calorimetric power and LHV, why didn't you do HHV?

The bomb calorimeter was used to determine the HHV of the material(s). It was also used as pre-treatment analysis to determine the Cl content of the material(s). The output value of the bomb calorimeter was HHV. The LHV was calculated starting from the HHV and H and moisture content.

  • You wrote that you did EDX in the experimental part, which usually goes with SEM, but I didn't see the results

EDX was used without SEM. The equipment for EDX analysis was performed by Shimadzu EDX 7000, which is not equipped with SEM.

  • You presented BET results and did not provide isotherms or pore distribution

We have added to the text a picture reporting the pore distribution of the three biocoals.

  • Why didn't you do FTIR of the solid residue to see the chemistry of the surface and functional groups of the solid residue?

We recognize that the FTIR analysis is useful for the determination of functional groups of carbon-based solids, as functional groups influence their absorption capacity. However, the use of FTIR for the determination of the functional groups was considered not relevant for the final scope of the proposed work. In fact, in the specific case of the biocoals obtained in this study, the carbon content represented a percentage between 25 and 28% of the solid, while the main fraction consisted of inorganic salts, mainly silica and alumina. Besides, it was decided to consider the solid as an inorganic adsorbent, and to determine the surface area.

  • You could do ICP to see the content of heavy metals

With the available instruments, ICP analysis can be applied only to liquid samples. In the case of solid materials, the sample requires to be pre-treated by an acidic digestion to convert it into a liquid sample. In the case of solid samples, the amount of silicon determined by ICP was found to be underestimated, due to the high concentration of silica and to the reduced efficacy of the pre-treatment of the solid samples.

[1]A. Salimbeni, A. M. Rizzo, and D. Chiaramonti, “Integration of two-stage thermochemical treatment and chemical leaching for extraction of advanced biochar and high value critical raw materials from sewage sludge,” E3S Web of Conferences, vol. 238, no. March 2019, 2021, doi: 10.1051/e3sconf/202123801008.

2 L. Doyle et al., Industrial Scale Hydrothermal Carbonization: new applications for wet biomass waste, 2016. Editor: Bárbara De Mena; Lucía Doyke; Michael Renz; Andrea Salimbeni. ISBN: 978-3-00-052950-4

Reviewer 2 Report

“Opportunities of integrating slow pyrolysis and chemical leaching for the extraction of critical raw materials from sewage sludge” is a research paper focused on the recovery of raw materials from biological sludge, the main residue of wastewater treatment. The work is well done but I suggest major amendments before publication.

·     In my opinion, different subsections of the Introduction should be merged to create a single section. I suggest also to shortened it focusing on main problems, background aspects and novelty aspects of your findings.

·      In the manuscript you have 8 Figures and 15 Tables. Are you sure that all are necessary? To have a more fluid text, I suggest to move not strictly necessary tables and figures (e.g., Figure 5,6, and 7) in the Supplementary materials. Moreover, info should be reported once in the text OR in the manuscript (e.g., Figure 2, 3, and 4 repeat some info already reported in the text). Are you sure that you could not create a single figure with three subfigures for them?

·     I suggest to integrate “Discussion” with more comparison with results already obtained in literature for similar materials.

·     In my opinion, conclusions should be shortened and focused on the main results of the work (e.g., Lines can be summarized, in my opinion). To stress the novelty of your work, I suggest to insert also in the conclusions who can be the possible stakeholder for your results. Scientific community? Technicians?

Author Response

The authors gratefully acknowledge the anonymous reviewer for its report, which offers the opportunity to better clarify on the aim and scope of the paper and the methodology adopted. Answers to all the reviewer’s questions and comments have been addressed, and reported below (answers text in italic).

Comment 1 -  In my opinion, different subsections of the Introduction should be merged to create a single section. I suggest also to shortened it focusing on main problems, background aspects and novelty aspects of your findings.

Answer: Authors thank for the suggestion. The introduction has been revised and sections have been merged according to the suggestion

  • In the manuscript you have 8 Figures and 15 Tables. Are you sure that all are necessary? To have a more fluid text, I suggest to move not strictly necessary tables and figures (e.g., Figure 5,6, and 7) in the Supplementary materials. Moreover, info should be reported once in the text OR in the manuscript (e.g., Figure 2, 3, and 4 repeat some info already reported in the text). Are you sure that you could not create a single figure with three subfigures for them?

Answer: Authors thank for the suggestion. Figure 2, 3, 4, 5, 6 have been moved to the “Supplementary materials” file, and a single figure merging Figure 2, 3 and 4 has been added to the text.

  • I suggest to integrate “Discussion” with more comparison with results already obtained in literature for similar materials.

Answer:  Discussion section has been modified accordingly. Comparison with other studies and related references have been added.

  • In my opinion, conclusions should be shortened and focused on the main results of the work (e.g., Lines can be summarized, in my opinion). To stress the novelty of your work, I suggest to insert also in the conclusions who can be the possible stakeholder for your results. Scientific community? Technicians?

Answer:  Conclusion section has been shortened. Authors thank for the suggestion; the potential stakeholders of the proposed process have been mentioned in the conclusion.

Reviewer 3 Report

Manuscript ID: water-2257187

Title: Opportunities of integrating slow pyrolysis and chemical leaching for the extraction of critical raw materials from sewage sludge

Journal: Water

Comment 1: "A new circular approach towards waste management, also promoted by the Circular Economy Package, and aimed at the transition to a circular economy [7], suggests that new routes for sludge utilization must be found, aimed at the recovery of its most valuable compounds, with environmental and economic benefits" Connect the characterization of SS and stability to the application of circular bioeconomy.

Comment 2: "…pathogenic microorganisms…" The suspended aggregates of xenic and axenic MOs must be considered and linked to the elution of inorganic elements.

Comment 3: "and of the ethanol-washed precipitate" Clear if you used sieving as a pretreatment.

Comment 4: "A more interesting strategy could be represented by the application of the leached biocoal as precursor for adsorbents production, or as SiO2 source in the construction industry, or in metallurgy, to produce silicon metal in submerged arc furnaces" Does this include VOCs adsorption?

Comment 5: "concurrently stabilizing organic matter end facilitating the recovery of valuable elements (P, C, etc.)" Study the metal stabilization mechanisms.

Comment 6: "2.4 Methodology of chemical leaching experiments" The experimental section must be combined to the porosity, phenomenographical, macroscopic and microscopic mechanisms, sensitivity and predictive analysis.

Comment 7: "The sewage sludge object of the study is produced by the secondary sedimentation phase after the biological treatment, then it is thickened, anaerobically digested and finally dewatered" and "However, poor literature studies are available on chemical leaching of biochar from sludge slow pyrolysis; moreover, an assessment of chemical leaching efficiency should be provided considering not only the extraction of phosphorus, but also of other inorganic elements, and the whole ash removal efficiency" 1- Define the Fe content (involve the role of Fe species) in sludge under the working temperature. 2- Assess the enhancement of sludge dewaterability. 3- The mathematical modelling of slow pyrolysis of sludge should be perfomed after the optimization of process conditions to maximize the yield. 4- Perform and discuss the findings of the multiple nonlinear regression, artificial neural network, and computational thermodynamics.

Comment 8: "The precipitation by KOH enabled to recover 100 % Al and P in a promising solid N-P-K inorganic fertiliser" You are asked to perform Monte Carlo study of the precipitation kinetics by KOH.

Comment 9: "Once phosphorus and other harmful elements (Na, S) are removed and valorized separately, the biocoal can be used in a set of industrial sectors, in particular: cement industry, and silicon production industry" Investigate the volrisation of secondary resources involving their palletisation.

Comment 10: "A chemical leaching experimental campaign was conducted at laboratory scale to extract and separate the desired inorganic elements from the pyrolyzed sludge" and "Using KOH determines the presence of K in the precipitate, increasing the value of the recovered compound in the fertilizers sector, since K is a key soil macronutrient" I see that you ignored in your paper the types and contamination loads (Inorganic mattters (and their transformation) as As, Cr, Cu, Fe, B, and rare earth elements and Organics as PCP and PCDD/F) of soils which affect the presented outcomes including the initial release of inorganic/organic species during the pyrolysis stage and during thermal treatment of sewage sludge!

Comment 11: "Chemical leaching by alkalis, such as NaOH, KOH and Ca(OH)2, is effective on the extraction of silica, alumina, and clay-bearing minerals (which represent 60-90 % of the total coal mineral matter) producing hydrated alkali-bearing silicate, aluminate and aluminosilicate complexes" Why you ignored the organic complexes and the oxidation processes and their contribution of acidity produced from the oxygenation of ferrous iron?

Comment 12: "Inorganic and organic sulphur may be extracted by NaOH or KOH", "Ultimate analysis allowed to determine carbon, hydrogen, nitrogen, sulphur and chlorine contents in the material, and it was conducted by a CHN-S analyser (LECO 175 TruSpec CHN-S) and by the use of a bomb calorimeter (LECO AC500), for pre-treatment, and a ion chromatography system (Metrohm 883 Basic IC plus) to determine chlorine", and "Compared to combustion process, it has been reported that pyrolysis processing generates less air pollutants emissions" 1- The chemical speciation and leaching behavior of emissions including hazardous trace elements must be cleared in the text. 2- Perform the batch pyrolysis and compare the results with combustion experiments, and then study sulphur transformations during thermal conversion.

Comment 13: "The use of aluminium polychloride (around 270 t/y) and iron chloride (approximately 10 t/y) in the plant as reagents results in a high inorganic matter content in the produced sludge" This sentence must come after the determination of the range of capacity between in t/d and t/y and the chloride in fertilizers of high N content and connected to "After the leaching test, to recover an inorganic compound of high quality for fertilizers production, a 15 % KOH solution was used for the chemical precipitation step after the leaching test" and P-recovery (consider the P bound to amphoteric aluminum and iron, and the little residual carbon) from sludge.

Comment 14: "The challenge is to maximize the extraction of phosphorus, but also to optimize the extraction of inorganic compounds, aiming to recover two products: the inorganic fraction and the upgraded char (a “biocoal”), both usable as end-of-waste products" and "Concerning case 2, an interesting surface area value, namely 70 m2/g, was reached, which is higher than typical waste material values and comparable to that shown by biochar produced from some lignocellulosic materials" 1- But lignocellulose requires too much quantity of reagents and is associated with unsatisfying disposal of polychlorinated compounds! 2- The pyrolysis of lignocellulosic materials/cellulose, lipids and liposoluble compounds, and the associated reductant atmospheres must be studied including the aromatic solvents, reactivity, product yield, and char morphology.

Comment 15: Table 2, Fig.7 and "Combined recovery of carbon and inorganic elements has been tested, with the main aim of phosphorous recovery, on hydrochar, the solid product generated by hydrothermal carbonization (HTC) [19]. By acid leaching with HCl and subsequent precipitation with NaOH, up to 71 % of the phosphorous contained in the hydrochar from sewage sludge HTC was recovered [20], while the use of HNO3 as leaching agent led to the recovery of 78 % of the phosphorous retained in the hydrochar from an Italian sewage sludge" Provide the optimum working conditions and the basic results of feedstock-dependent P-recovery in the slow pilot unit scale.

Comment 16: Tables 3 and 9 – Investigate the fate and distribution of HMs during thermal processing of this sewage sludge.

Comment 17: Table 8 and "When the pyrolysis process is applied to maximize the production of char, thus the recovery of carbon in the solid product, it is commonly known as “slow pyrolysis”" Connect them to the charcterisation of slow pyrolysis products (the product yield and characterisation of the liquid and solid fractions must be included) from segregated wastes. Analyse the energy production and recovery.

Comment 18: Fig.8, "Then, a precipitation of the eluate from the alkali leaching step, performed by CaCl2, enables the separated recovery of P as calcium phosphate, leaving Al dissolved in the remaining liquid, with a potential of application in wastewater treatment plants. In fact, while Ca should react with P to form calcium phosphate, Cl and Al remain in the solution and can be used to obtain aluminium polychloride", "The use of HNO3 produced a N-rich solution and opens the possibility to apply a precipitation on the eluate from the first step by KOH or Ca(OH)2, in the view of recovering the dissolved Ca as calcium nitrate, which could be applied for fertilizers production", "In fact, according to [22], the P and Al present as aluminium phosphate can be dissolved in alkaline conditions. Then, a precipitation of the eluate from the alkali leaching step, performed by CaCl2, enables the separated recovery of P as calcium phosphate, leaving Al dissolved in the remaining liquid, with a potential of application in wastewater treatment plants. In fact, while Ca should react with P to form calcium phosphate, Cl and Al remain in the solution and can be used to obtain aluminium polychloride", and "In addition, eluate Al, P and Ca are totally recovered in solid form. The negative aspects of this acid leaching process are related to the simultaneous extraction of Ca, P and Al" 1- Perform the chemical recycling of the waste material produced (consider the particle size ranges and the pyrolysis gas- CaO phase behavior through acid leaching) through pyrolysis. 2- Assess the physiochemical properties of the derived products.

Comment 19: "Due to the presence of aluminium, which reduces the quality of the recovered inorganic compound" More clarifications are requested.

Comment 20: "Silica extraction by alkali leaching could be performed, but at more severe conditions" Elucidate the influencing parameters on laboratory performance testing and present and discuss the alkali-silica reactions.

Comment 21: "A more interesting strategy could be represented by the application of the leached biocoal as precursor for adsorbents production, or as SiO2 source in the construction industry, or in metallurgy, to produce silicon metal in submerged arc furnaces" 1- The power in megavolt-amperes must be ranged. 2- The exergy analysis of Si metallurgy should be performed.

Comment 22: "The reduction reaction takes place by adding a reducing agent to the silica" 1- This is related to the contents of impurities in silica, temperature, reactive blend condition on the thermal properties, type of coal, unfavorable addition of excess silicon. 2- Which reducing silicon-based agents you mean as can we use H2 or phenol or was it suitable to add Rhenium-catalyzed deoxygenation of epoxides? (Mention the redox reaction occurs involving the oxidation of surface)

Comment 23: The language should be revised.

Comment 24: References:

Note 1: "Due to the high ash content and to the reduced calorific value of the dry sludge, incineration represents an expensive disposal system for sludge producers" Add the following references:

2012. Sewage sludge pyrolysis for liquid production: A review. Renewable and Sustainable Energy Reviews 16(5), 2781-2805.

2022.Co-pyrolysis of paper mill sludge and textile dyeing sludge with high calorific value solid waste: Pyrolysis kinetics, products distribution, and pollutants transformation. Fuel 329, 125433.

Note 2: "The pyrolysis process always generates a solid carbonaceous matrix (char), a mixture of condensable gases (water and organic compounds) and a mixture of permanent gases (CO, CO2, CH4, H2, etc.)" Add the following references:

2022. Comprehensive review on mechanism analysis and numerical simulation of municipal solid waste incineration process based on mechanical grate. Fuel 320, 123826.

2023. Investigation on dioxins emission characteristic during complete maintenance operating period of municipal solid waste incineration. Environmental Pollution 318, 120949.

Note 3: "Chemical leaching is a process which allows to separate the soluble components of a solid material by dissolving them in a liquid phase" Add the following reference:

2022.Pilot-Plant Investigation of the Leaching Process for the Recovery of Scandium from Red Mud. Industrial & Engineering Chemistry Research 41(23), 5794-5801.

Note 4: "Slow pyrolysis process enables to convert dry sludge into a porous solid with carbon in stable form, which facilitates the leaching process efficacy" Add the following reference:

2015. A comparative review of biochar and hydrochar in terms of production, physico-chemical properties and applications. Renewable and Sustainable Energy Reviews 45, 359-378.

Note 5: "To make char from sewage sludge usable, chemical leaching of sewage sludge char is an effective system to reduce char ash content and to extract valuable mineral elements (such as phosphorous) in the view of their recovery" Add the following reference:

2020.Biochar – Recovery Material from Pyrolysis of Sewage Sludge: A Review. Waste and Biomass Valorization 11(7), 3677-3709.

Note 6: "Char from sludge pyrolysis is a carbon-rich material, hydrophobic, with low volatile content and relevant porosity" Add the following reference:

2020.Biochar physicochemical properties: pyrolysis temperature and feedstock kind effects. Reviews in Environmental Science and Bio/Technology 19(1), 191-215.

Note 7: "Since most of the carbon contained in char is recalcitrant, its use on soil might allow to sequester carbon" Add the following reference:

2014. Characteristics of biochars from crop residues: Potential for carbon sequestration and soil amendment. Journal of Environmental Management 146, 189-197.

Note 8: "In fact, during pyrolysis, part of the feedstock organic matter is volatilized into pyrogas, while ashes are inert to the process and remain in the solid product" Add the following references:

2022. Dioxin emission prediction based on improved deep forest regression for municipal solid waste incineration process. Chemosphere 294, 133716.

2022. Prediction of dioxin emission from municipal solid waste incineration based on expansion, interpolation, and selection for small samples. Journal of Environmental Chemical Engineering10(5), 108314.

Note 9: "Moreover, due to its high ash content and thus low calorific value, char from sludge slow pyrolysis…" Add the following reference:

2016. Energy conversion assessment of vacuum, slow and fast pyrolysis processes for low and high ash paper waste sludge. Energy Conversion and Management 111, 103-114.

Note 10: "However, the high concentration of both inorganic elements and carbon make sludge-derived char an interesting raw material, usable as a source of nutrients, and renewable carbon" Add the following reference:

2022.Carbon-based material derived from biomass waste for wastewater treatment. Environmental Advances 9, 100259.

Note 11: "As visible from Table 14, the surface area increases with the decrease of ash content in the biocoal" Add the following reference:

2017. Production and characterization of smokeless bio-coal briquettes incorporating plastic waste materials. Environmental Technology & Innovation 8, 233-245.

Note 12: "During pyrolysis, part of the organic matter is devolatilised and converted into pyrogas, determining the increase of the inorganic compounds (ash) percentage content" Add the following reference:

2023.Sustainable Valorisation of Animal Manures via Thermochemical Conversion Technologies: An Inclusive Review on Recent Trends. Waste and Biomass Valorization 14(2), 553-582.

Note 13: "Quartz sand and carbon are fed in appropriate proportions through the top, and liquid silicon is extracted at the bottom" Add the following book chapter:

2008. Production of metallurgical-grade silicon in an electric arc furnace. SGTE Casebook, pp.415-424.

Note 14: "…eluate were dosed to have a Ca:P molar ratio of 1.5, optimal for calcium phosphate recovery as suggested by literature" Add the following references:

2017. Hydroxyapatite and Fluoroapatite behavior with pH Change. International Medical Journal 24(5), 407-410.

2017. A descriptive study – in vitro: new validated method for checking Hap and Fap Behaviours. International Medical Journal 24(5), 394-397.

Comment 25: The Statistical Analysis is mandatory for publishing the current paper.

Author Response

The authors gratefully acknowledge the anonymous reviewer for its report, which offers the opportunity to better clarify on the aim and scope of the paper and the methodology adopted. Answers to all the reviewer’s questions and comments have been addressed, and reported below (answers text in italic).

Comment 1: "A new circular approach towards waste management, also promoted by the Circular Economy Package, and aimed at the transition to a circular economy [7], suggests that new routes for sludge utilization must be found, aimed at the recovery of its most valuable compounds, with environmental and economic benefits" Connect the characterization of SS and stability to the application of circular bioeconomy.

Answer: authors thank for the suggestion. The Introduction has been integrated with new text to address the requirements.

Comment 2: "…pathogenic microorganisms…" The suspended aggregates of xenic and axenic MOs must be considered and linked to the elution of inorganic elements.

Answer: despite the topic is interesting, the authors could not have the adequate instruments to perform such laboratory analysis. This activity could be performed in next studies.

Comment 3: "and of the ethanol-washed precipitate" Clear if you used sieving as a pretreatment.

Answer: No sieving was performed before washing treatment. The precipitate was washed during filtration on a büchner funnel.

Comment 4: "A more interesting strategy could be represented by the application of the leached biocoal as precursor for adsorbents production, or as SiO2 source in the construction industry, or in metallurgy, to produce silicon metal in submerged arc furnaces" Does this include VOCs adsorption?

Answer: The adsorption of VOCs is one of the potential applications of biocoal. This research activity is under investigation and could be published in a new manuscript.

Comment 5: "concurrently stabilizing organic matter end facilitating the recovery of valuable elements (P, C, etc.)" Study the metal stabilization mechanisms.

Answer: this study will be object of a future work, to be focused on testing the biocoal as adsorbent.

Comment 6: "2.4 Methodology of chemical leaching experiments" The experimental section must be combined to the porosity, phenomenographical, macroscopic and microscopic mechanisms, sensitivity and predictive analysis.

Answer: as regards to porosity, the pore distribution of the biocoals was added in the manuscript. Concerning the other mentioned analysis, authors are still studying some aspects of the processes.

Comment 7: "The sewage sludge object of the study is produced by the secondary sedimentation phase after the biological treatment, then it is thickened, anaerobically digested and finally dewatered" and "However, poor literature studies are available on chemical leaching of biochar from sludge slow pyrolysis; moreover, an assessment of chemical leaching efficiency should be provided considering not only the extraction of phosphorus, but also of other inorganic elements, and the whole ash removal efficiency"

  • Define the Fe content (involve the role of Fe species) in sludge under the working temperature.

Answer: as part of the study, the Fe content of sewage sludge and char was determined analytically.

  • Assess the enhancement of sludge dewaterability.

Answer: The enhancement of sludge dewaterability was not assesses in this study, the authors focused on other characteristics of the material.

  • The mathematical modelling of slow pyrolysis of sludge should be performed after the optimization of process conditions to maximize the yield.

Answer: The elaboration of a mathematical model is programmed and will be hopefully applied as suggested by the revisor (among other applications).

  • Perform and discuss the findings of the multiple nonlinear regression, artificial neural network, and computational thermodynamics.

Answer: Authors do not consider this suggestion in line with the scope of the work.

Comment 8: "The precipitation by KOH enabled to recover 100 % Al and P in a promising solid N-P-K inorganic fertiliser" You are asked to perform Monte Carlo study of the precipitation kinetics by KOH.

Answer: The proposal is very interesting. A Monte Carlo study could be performed as a follow up study of the proposed work.

Comment 9: "Once phosphorus and other harmful elements (Na, S) are removed and valorized separately, the biocoal can be used in a set of industrial sectors, in particular: cement industry, and silicon production industry" Investigate the volrisation of secondary resources involving their palletisation.

Answer: Thanks for the suggestion, this investigation will be performed during a study dedicated to this specific activity.

Comment 10: "A chemical leaching experimental campaign was conducted at laboratory scale to extract and separate the desired inorganic elements from the pyrolyzed sludge" and "Using KOH determines the presence of K in the precipitate, increasing the value of the recovered compound in the fertilizers sector, since K is a key soil macronutrient" I see that you ignored in your paper the types and contamination loads (Inorganic mattters (and their transformation) as As, Cr, Cu, Fe, B, and rare earth elements and Organics as PCP and PCDD/F) of soils which affect the presented outcomes including the initial release of inorganic/organic species during the pyrolysis stage and during thermal treatment of sewage sludge!

Answer: The authors recognize the interest of analysing the behaviour of metals and organics during sludge thermal treatment. About Cr, Cu, Fe, B, their concentration in both the solids and liquid products has been determined. About the production of PCP and PCDD/F, the concentration in the pyrolysis gases was not investigated due to a lack of adequate instrument. The applicability of the inorganic fertiliser in the soil, a specific work on assessing the performances and contamination impact, is planned as follow up of this study.

Comment 11: "Chemical leaching by alkalis, such as NaOH, KOH and Ca(OH)2, is effective on the extraction of silica, alumina, and clay-bearing minerals (which represent 60-90 % of the total coal mineral matter) producing hydrated alkali-bearing silicate, aluminate and aluminosilicate complexes" Why you ignored the organic complexes and the oxidation processes and their contribution of acidity produced from the oxygenation of ferrous iron?

Answer: authors thank the revisor for the suggestion, this aspect will be taken in account in next activities concerning alkali-acid leaching processes.

Comment 12: "Inorganic and organic sulphur may be extracted by NaOH or KOH", "Ultimate analysis allowed to determine carbon, hydrogen, nitrogen, sulphur and chlorine contents in the material, and it was conducted by a CHN-S analyser (LECO 175 TruSpec CHN-S) and by the use of a bomb calorimeter (LECO AC500), for pre-treatment, and a ion chromatography system (Metrohm 883 Basic IC plus) to determine chlorine", and "Compared to combustion process, it has been reported that pyrolysis processing generates less air pollutants emissions"

  • The chemical speciation and leaching behavior of emissions including hazardous trace elements must be cleared in the text.

Answer: Thanks for the comment. Analysis of the emissions during laboratory leaching tests could not be performed at the time of the experiment. Emissions will be analysed in a new study performed in a chemical leaching pilot plant.

  • Perform the batch pyrolysis and compare the results with combustion experiments, and then study sulphur transformations during thermal conversion.

Answer: the suggestion is interesting. Sulphur transformation study will be considered in future experiments.

Comment 13: "The use of aluminium polychloride (around 270 t/y) and iron chloride (approximately 10 t/y) in the plant as reagents results in a high inorganic matter content in the produced sludge" This sentence must come after the determination of the range of capacity between in t/d and t/y and the chloride in fertilizers of high N content and connected to "After the leaching test, to recover an inorganic compound of high quality for fertilizers production, a 15 % KOH solution was used for the chemical precipitation step after the leaching test" and P-recovery (consider the P bound to amphoteric aluminium and iron, and the little residual carbon) from sludge.

Answer: the daily consumption of aluminium polychloride and iron chloride has been added to the manuscript. About the chlorine concentration (% db) in the raw sludge, and in the char, this value has been reported.

Comment 14: "The challenge is to maximize the extraction of phosphorus, but also to optimize the extraction of inorganic compounds, aiming to recover two products: the inorganic fraction and the upgraded char (a “biocoal”), both usable as end-of-waste products" and "Concerning case 2, an interesting surface area value, namely 70 m2/g, was reached, which is higher than typical waste material values and comparable to that shown by biochar produced from some lignocellulosic materials"

  • But lignocellulose requires too much quantity of reagents and is associated with unsatisfying disposal of polychlorinated compounds!
  • The pyrolysis of lignocellulosic materials/cellulose, lipids and liposoluble compounds, and the associated reductant atmospheres must be studied including the aromatic solvents, reactivity, product yield, and char morphology.

Answer: authors thank for the comment. The pyrolysis of lignocellulosic biomass was not in the scope of the work. This topic could be studied in a future work.

Comment 15: Table 2, Fig.7 and "Combined recovery of carbon and inorganic elements has been tested, with the main aim of phosphorous recovery, on hydrochar, the solid product generated by hydrothermal carbonization (HTC) [19]. By acid leaching with HCl and subsequent precipitation with NaOH, up to 71 % of the phosphorous contained in the hydrochar from sewage sludge HTC was recovered [20], while the use of HNO3 as leaching agent led to the recovery of 78 % of the phosphorous retained in the hydrochar from an Italian sewage sludge". Provide the optimum working conditions and the basic results of feedstock-dependent P-recovery in the slow pilot unit scale.

Answer: This comment fits the interest of the authors about the proposed integrated process. As shown in the paper, full recovery of phosphorus was achieved. A specific study is planned to determine the optimum process condition to be adopted in the pilot plant for maximizing the whole process efficiency. Data could be presented in a new manuscript.

Comment 16: Tables 3 and 9

Investigate the fate and distribution of HMs during thermal processing of this sewage sludge.

Answer: The increase of HMs in the solid after the pyrolysis process has been mentioned in the manuscript.

Comment 17: Table 8 and "When the pyrolysis process is applied to maximize the production of char, thus the recovery of carbon in the solid product, it is commonly known as “slow pyrolysis”. Connect them to the charcterisation of slow pyrolysis products (the product yield and characterisation of the liquid and solid fractions must be included) from segregated wastes. Analyse the energy production and recovery.

Answer: The process energy balance and liquids characterization has been added to the manuscript.

Comment 18: Fig.8, "Then, a precipitation of the eluate from the alkali leaching step, performed by CaCl2, enables the separated recovery of P as calcium phosphate, leaving Al dissolved in the remaining liquid, with a potential of application in wastewater treatment plants. In fact, while Ca should react with P to form calcium phosphate, Cl and Al remain in the solution and can be used to obtain aluminium polychloride", "The use of HNO3 produced a N-rich solution and opens the possibility to apply a precipitation on the eluate from the first step by KOH or Ca(OH)2, in the view of recovering the dissolved Ca as calcium nitrate, which could be applied for fertilizers production", "In fact, according to [22], the P and Al present as aluminium phosphate can be dissolved in alkaline conditions. Then, a precipitation of the eluate from the alkali leaching step, performed by CaCl2, enables the separated recovery of P as calcium phosphate, leaving Al dissolved in the remaining liquid, with a potential of application in wastewater treatment plants. In fact, while Ca should react with P to form calcium phosphate, Cl and Al remain in the solution and can be used to obtain aluminium polychloride", and "In addition, eluate Al, P and Ca are totally recovered in solid form. The negative aspects of this acid leaching process are related to the simultaneous extraction of Ca, P and Al"

  • Perform the chemical recycling of the waste material produced (consider the particle size ranges and the pyrolysis gas- CaO phase behavior through acid leaching) through pyrolysis.
  • Assess the physiochemical properties of the derived products.

Answer: The authors thank the revisor for the suggestion. The target of the work described in the manuscript was to recycle the carbon and the inorganic elements contained in the produced char. Additional chemical recycling solutions will be tested concerning pyrolysis gases and liquid in future experiments.

Comment 19: "Due to the presence of aluminium, which reduces the quality of the recovered inorganic compound" More clarifications are requested.

Answer: Aluminium is not considered harmful for soil, but its presence lowers the concentration of other useful elements for soil (NPK), leading the quality of the fertilizing product to decrease. This consideration has been added to the text. Agronomic experiments on the obtained inorganic compound will be performed in the future.

Comment 20: "Silica extraction by alkali leaching could be performed, but at more severe conditions" Elucidate the influencing parameters on laboratory performance testing and present and discuss the alkali-silica reactions.

Answer: silica extraction by alkali leaching was not performed in the presented work. About the alkali leaching process to be adopted, reference is reported in the text[1]: Alkali-silica reactions have not been investigated in the study, but it could be object of future works.

Comment 21: "A more interesting strategy could be represented by the application of the leached biocoal as precursor for adsorbents production, or as SiO2 source in the construction industry, or in metallurgy, to produce silicon metal in submerged arc furnaces"

  • The power in megavolt-amperesmust be ranged.
  • The exergy analysis of Si metallurgy should be performed.

Answer: as previously mentioned, this research activity is under investigation and could be published in a new manuscript. Exergy analysis of Si metallurgy by using the obtained products could be part of a future work.

Comment 22: "The reduction reaction takes place by adding a reducing agent to the silica"

  • This is related to the contents of impurities in silica, temperature, reactive blend condition on the thermal properties, type of coal,unfavorable addition of excess silicon.
  • Which reducing silicon-based agents you mean as can we use H2 or phenol or was it suitable to add Rhenium-catalyzed deoxygenation of epoxides? (Mention the redox reaction occurs involving the oxidation of surface)

Answer: This topic is of great interest for the authors. The target of the authors is to test the application of obtained biocoals as reducing agents for silicon metal production, replacing commonly used reducing agents, which consist in carbon in the form of mineral carbon, or charcoal or wood-chips[2]. The results will be described in future works. Reduction reaction which take place  in silicon metal production are well descrived by Abolpour B et al, 2020[3].

Comment 23: The language should be revised.

Answer: the language of the manuscript has been revised. Some paragraphs have been re-written.

Comment 24: References:

Note 1: "Due to the high ash content and to the reduced calorific value of the dry sludge, incineration represents an expensive disposal system for sludge producers" Add the following references:

  1. Sewage sludge pyrolysis for liquid production: A review. Renewable and Sustainable Energy Reviews 16(5), 2781-2805.

2022.Co-pyrolysis of paper mill sludge and textile dyeing sludge with high calorific value solid waste: Pyrolysis kinetics, products distribution, and pollutants transformation. Fuel 329, 125433.

Note 2: "The pyrolysis process always generates a solid carbonaceous matrix (char), a mixture of condensable gases (water and organic compounds) and a mixture of permanent gases (CO, CO2, CH4, H2, etc.)" Add the following references:

  1. Comprehensive review on mechanism analysis and numerical simulation of municipal solid waste incineration process based on mechanical grate. Fuel 320, 123826.
  2. Investigation on dioxins emission characteristic during complete maintenance operating period of municipal solid waste incineration. Environmental Pollution 318, 120949.

Note 3: "Chemical leaching is a process which allows to separate the soluble components of a solid material by dissolving them in a liquid phase" Add the following reference:

2022.Pilot-Plant Investigation of the Leaching Process for the Recovery of Scandium from Red Mud. Industrial & Engineering Chemistry Research 41(23), 5794-5801.

Note 4: "Slow pyrolysis process enables to convert dry sludge into a porous solid with carbon in stable form, which facilitates the leaching process efficacy" Add the following reference:

  1. A comparative review of biochar and hydrochar in terms of production, physico-chemical properties and applications. Renewable and Sustainable Energy Reviews 45, 359-378.

Note 5: "To make char from sewage sludge usable, chemical leaching of sewage sludge char is an effective system to reduce char ash content and to extract valuable mineral elements (such as phosphorous) in the view of their recovery" Add the following reference:

2020.Biochar – Recovery Material from Pyrolysis of Sewage Sludge: A Review. Waste and Biomass Valorization 11(7), 3677-3709.

Note 6: "Char from sludge pyrolysis is a carbon-rich material, hydrophobic, with low volatile content and relevant porosity" Add the following reference:

2020.Biochar physicochemical properties: pyrolysis temperature and feedstock kind effects. Reviews in Environmental Science and Bio/Technology 19(1), 191-215.

Note 7: "Since most of the carbon contained in char is recalcitrant, its use on soil might allow to sequester carbon" Add the following reference:

  1. Characteristics of biochars from crop residues: Potential for carbon sequestration and soil amendment. Journal of Environmental Management 146, 189-197.

Note 8: "In fact, during pyrolysis, part of the feedstock organic matter is volatilized into pyrogas, while ashes are inert to the process and remain in the solid product" Add the following references:

  1. Dioxin emission prediction based on improved deep forest regression for municipal solid waste incineration process. Chemosphere 294, 133716.
  2. Prediction of dioxin emission from municipal solid waste incineration based on expansion, interpolation, and selection for small samples. Journal of Environmental Chemical Engineering10(5), 108314.

Note 9: "Moreover, due to its high ash content and thus low calorific value, char from sludge slow pyrolysis…" Add the following reference:

  1. Energy conversion assessment of vacuum, slow and fast pyrolysis processes for low and high ash paper waste sludge. Energy Conversion and Management 111, 103-114.

Note 10: "However, the high concentration of both inorganic elements and carbon make sludge-derived char an interesting raw material, usable as a source of nutrients, and renewable carbon" Add the following reference:

2022.Carbon-based material derived from biomass waste for wastewater treatment. Environmental Advances 9, 100259.

Note 11: "As visible from Table 14, the surface area increases with the decrease of ash content in the biocoal" Add the following reference:

  1. Production and characterization of smokeless bio-coal briquettes incorporating plastic waste materials. Environmental Technology & Innovation 8, 233-245.

Note 12: "During pyrolysis, part of the organic matter is devolatilised and converted into pyrogas, determining the increase of the inorganic compounds (ash) percentage content" Add the following reference:

2023.Sustainable Valorisation of Animal Manures via Thermochemical Conversion Technologies: An Inclusive Review on Recent Trends. Waste and Biomass Valorization 14(2), 553-582.

Note 13: "Quartz sand and carbon are fed in appropriate proportions through the top, and liquid silicon is extracted at the bottom" Add the following book chapter:

  1. Production of metallurgical-grade silicon in an electric arc furnace. SGTE Casebook, pp.415-424.

Note 14: "…eluate were dosed to have a Ca:P molar ratio of 1.5, optimal for calcium phosphate recovery as suggested by literature" Add the following references:

  1. Hydroxyapatite and Fluoroapatite behavior with pH Change.International Medical Journal 24(5), 407-410.
  2. A descriptive study – in vitro: new validated method for checking Hap and Fap Behaviours. International Medical Journal24(5),394-397.

Answer: Authors thank for the suggestion. The suggested references have been added to the manuscript, with the exception of the following:

Note 2: 2023. Investigation on dioxins emission characteristic during complete maintenance operating period of municipal solid waste incineration. Environmental Pollution 318, 120949.

The suggested reference is very interesting, but refers to incineration, thus just the first suggested reference has been added to the manuscript.

Note 8: 2022. Dioxin emission prediction based on improved deep forest regression for municipal solid waste incineration process. Chemosphere 294, 133716

  1. Prediction of dioxin emission from municipal solid waste incineration based on expansion, interpolation, and selection for small samples. Journal of Environmental Chemical Engineering10(5), 108314.

The suggested references are very interesting, but very specific on dioxin emission from municipal solid waste incineration, not mentioning pyrolysis.

Comment 25: The Statistical Analysis is mandatory for publishing the current paper.

Answer: Authors thank for the suggestion. A statistical analysis has been added to the manuscript

[1] B. Mohanty, R., Mishra, S.K., Mohapatra, S.S., Choudhury, S., Pattanayak, “Extraction of Silica (Sio2) from Coal Fly Ash by Leaching and Sintering Technology. In: Mahanta, P., Kalita, P., Paul, A., Banerjee, A. (eds) Advances in Thermofluids and Renewable Energy. Lecture Notes in Mechanical Engineering. Springer, Singapore.,” 2022, doi: https://doi.org/10.1007/978-981-16-3497-0_30.

[2] European Carbon and Graphite Association (ECGA), “Silicon metal production.” https://www.carbonandgraphite.org/index.php/silicon-metal-production (accessed Jul. 17, 2022).

[3] Abolpour B, Shamsoddini R. Mechanism of reaction of silica and carbon for producing silicon carbide. Progress in Reaction Kinetics and Mechanism. 2020;45. doi:10.1177/1468678319891416

Reviewer 4 Report

This paper attempts to produce biochar from sewage sludge at low temperatures and then recover useful components by solvent extraction or separate aluminum.

1)In Fig. 1, Char and Biocoal seem to have different shapes. Is it correct to think that this is due to the dissolution of the carbon content?

2)Elute is confusing,  you want to show that it's colored yellow?

3)I think it would be better to clarify the difference between Char and Biochar.  There are expressions for Char, Leached Char, and Biocoal, which is difficult to understand.

4)Recovery of components from Biocar generated by pyrolysis is an important issue, but there is no discussion of the gases generated. Is it just collecting liquids?

5)How should we think about recovery of potassium? Since we are adding KOH, the potassium concentration increases. Excess sludge is inherently low in potassium content, making it difficult to recover from sludge.

Author Response

The authors gratefully acknowledge the anonymous reviewer for its report, which offers the opportunity to better clarify on the aim and scope of the paper and the methodology adopted. Answers to all the reviewer’s questions and comments have been addressed, and reported below

This paper attempts to produce biochar from sewage sludge at low temperatures and then recover useful components by solvent extraction or separate aluminum.

  • 1)In Fig. 1, Char and Biocoal seem to have different shapes. Is it correct to think that this is due to the dissolution of the carbon content?

Authors confirm that the different shapes of char and biocoal are due to the effect of leaching process. However, the main effect of the leaching process is to dissolve inorganic compounds in the liquid. Therefore, the difference shapes of char and biocoal is considered as a consequence of two factors: the dissolution of inorganic elements from solid particles to the liquid and the increase of the carbon content in the bicoal, compared to the char.

  • 2)Elute is confusing, you want to show that it's colored yellow?

The authors just wanted to represent the eluate in the scheme, showing that the extraction of inorganic compounds from the char produces a coloured liquid.

  • 3)I think it would be better to clarify the difference between Char and Biochar.  There are expressions for Char, Leached Char, and Biocoal, which is difficult to understand.

The authors thank for the suggestion. A nomenclature has been added in the article before the introduction section. Moreover, some corrections have been included in the text to avoid confusion.

  • 4)Recovery of components from Biochar generated by pyrolysis is an important issue, but there is no discussion of the gases generated. Is it just collecting liquids?

Permanent gases generated by the pyrolysis process could not be analysed due to a lack of suitable gas analysis instruments. Condensable fraction of the pyrogas was collected, weighted and analysed in order to assess the mass distribution of three main pyrolysis products. The description of the procedure and the characterization of the condensable fraction has been added to the text.

  • 5)How should we think about recovery of potassium? Since we are adding KOH, the potassium concentration increases. Excess sludge is inherently low in potassium content, making it difficult to recover from sludge.

Potassium extraction efficiency could be determined only for case 1 and case 2. On the contrary, this value was not included for case 3, because of the use of KOH as leaching agent. The extraction efficiency of potassium for Case 1 and case 2 was calculated. This has been added in the article (lines 612 – 615).

Reviewer 5 Report

The paper presents an investigation of the integration of slow pyrolysis and chemical leaching for sludge treatment. The topic is interesting in terms of waste management and circular economy. The paper is quite long and complex. It can be seen that the Authors put the effort into their research, however, the manuscript could be better organized. My comments are:

  1. Why was a temperature of 710 C chosen in terms of ash characterization? 550 C is typical for biomass and can be applied to sewage sludge as well, but why 710 C? Please elaborate briefly.
  2. Line 398: char SPYRO - please introduce the term "SPYRO" earlier in the manuscript, before it appears in Table 5
  3. What about tar formation during the pyrolysis process? Did the Authors collect and analyzed any tar?
  4. The Discussion section contains some repetition from the Results section and does not introduce any novel thoughts. Please consider merging these two sections into one (Results and Discussion).
  5. The Conclusion section is too long. It should present the main findings of the work. The part concerning the possible use of obtained products in industries should be moved into the Results and Discussion. It can be only briefly summarized in the Conclusions. 
  6. Some editing errors:

Line 93 double dot

Line 334 and below: numbers in chemical formulas should be in subscripts

Author Response

The authors gratefully acknowledge the anonymous reviewer for its report, which offers the opportunity to better clarify on the aim and scope of the paper and the methodology adopted. Answers to all the reviewer’s questions and comments have been addressed, and reported below.

The paper presents an investigation of the integration of slow pyrolysis and chemical leaching for sludge treatment. The topic is interesting in terms of waste management and circular economy. The paper is quite long and complex. It can be seen that the Authors put the effort into their research, however, the manuscript could be better organized. My comments are:

  1. Why was a temperature of 710 C chosen in terms of ash characterization? 550 C is typical for biomass and can be applied to sewage sludge as well, but why 710 C? Please elaborate briefly.

Authors agree on that the typical ash determination for biomasses is usually prepared from the fuel by incinerating at 550 °C. In fact, this value is always reported in the tables related to the characterization of the processed and obtained materials.  Other incineration temperatures (710 °C or 815 °C) are used for solid minerals fuels. For this reason, given the aim to recover carbon as substitute of fossil coal, it was considered useful to determine ashes also at 710 °C.

  1. Line 398: char SPYRO - please introduce the term "SPYRO" earlier in the manuscript, before it appears in Table

Added to the text.

  1. What about tar formation during the pyrolysis process? Did the Authors collect and analyzed any tar?

The condensable fraction of pyrogas (also know as pyrolysis, bio-oil, TAR) was collected and weighted. Water was separated from the bio-oil. The bio-oil, , was characterized by water content, C, H, N and HHV. Data have been added to the paper. No specific TAR analysis has been performed.

  1. The Discussion section contains some repetition from the Results section and does not introduce any novel thoughts. Please consider merging these two sections into one (Results and Discussion).

The suggestion has been appreciated. The chapter “results” has been renamed with results and discussion”, sub-paragraphs have been added.

  1. The Conclusion section is too long. It should present the main findings of the work. The part concerning the possible use of obtained products in industries should be moved into the Results and Discussion. It can be only briefly summarized in the Conclusions. 

Conclusion has been modified by reducing the length and removing some paragraphs.

  1. Some editing errors:

Line 93 double dot

Line 334 and below: numbers in chemical formulas should be in subscripts

Editing errors have been fixed.

Round 2

Reviewer 1 Report

You have answered my questions adequately, therefore I give my opinion that the paper should be accepted in the journal.

Reviewer 2 Report

I thanks the authors for their improvements. In my opinion the work can now be published.

Reviewer 3 Report

Manuscript ID: water-2257187R1

Title: Opportunities of integrating slow pyrolysis and chemical leaching for the extraction of critical raw materials from sewage sludge

Journal: Water

The authors have adequately addressed most of my previous comments and suggestions. In addition, the revision is satisfactory and the changes are acceptable.